# Ultrathin organosiloxane membrane for precision organic solvent nanofiltration

Jihoon Choi[1,4], Keonwoo Choi[1,4], YongSung Kwon[2], Daehun Kim[1,2], Youngmin Yoo[2], Sung Gap Im ®[1,3] ✉ & Dong-Yeun Koh ®[1,3] ✉

Promising advances in membrane technology can lead to energy-saving and eco-friendly solutions in industrial sectors. This work demonstrates a highly selective membrane with ultrathin and highly interconnected organosiloxane polymer nanolayers by initiated chemical vapor deposition to effectively separate solutes within the molecular weight range of 150–300 g mol⁻¹. We optimize the poly(1,3,5,7-tetravinyl-1,3,5,7-tetramethylcyclotetrasiloxane) membrane by adjusting both the thickness of the selective layer and the pore sizes of its support membranes. Notably, the 29 nm selective layer imparts a uniformly narrow molecular sieving property, providing a record-high solute-solute selectivity of 39.88 for different-sized solutes. Furthermore, a solute-solute selectivity of 11.04 was demonstrated using the real-world active pharmaceutical ingredient mixture of Acyclovir and Valacyclovir, key components for Herpes virus treatment, despite their molecular weight difference of less than 100 g mol⁻¹. The highly interconnected membrane is expected to meet rigorous requirements for high-standard active pharmaceutical ingredient separation.

Membrane processes capable of differentiating molecules based on the size and shape of the molecules would consume far less energy than conventional thermal separations and could reduce the carbon intensity of the chemical industries. New mechanically and chemically robust molecular sieving membrane materials are indeed required to allow for membranes in harsh conditions involving complex organic solvents ubiquitously used in chemical, petrochemical, and pharmaceutical industries. The active pharmaceutical ingredient (API) industry, heavily dependent on organic solvents, is expected to grow exponentially due to the steep increase of chronic diseases in high global populations. Large volumes of organic solvents are utilized as reaction media and for recycling and reusing raw materials in the continuous, mass-production pharmaceutical industry[1,2]. Due to the toxic nature of solvents, however, waste disposal is also challenging, and reuse is also difficult due to the requirement of high purity of organic solvent[3,4].

Organic solvent nanofiltration (OSN), a cost-effective and environmentally friendly alternative for distillation or crystallization, allows for the effective separation of solute molecules with molecular weights (MWs) ranging from 100 to 2000 g mol⁻¹ in organic solvents[5]. Various polymers with rigid backbones with less swellability in organic solvents, such as poly(ether-ether-ketone), polybenzimidazole, and polymers of intrinsic microporosity (PIMs), were cast into a thin membrane film and evaluated for organic solvent permeation[6–8]. Most of the membrane materials reported to date commonly showed only limited applicability to the separation of solute molecules only >300 g mol⁻¹ from organic solvents (i.e., solvent−solute separations), represented with molecular weight cut-off (MWCO, solute size at 90% rejection via membrane) around 300 g mol⁻¹, due to the limited rigidity and high swellability of the active polymers. On the other hand, membrane processes for MWCO < 150 g mol⁻¹ is usually regarded as molecularly selective, solvent−solvent separation process and is

[1]Department of Chemical and Biomolecular Engineering (BK21 Four), Korea Advanced Institute of Science and Technology (KAIST), 291 Daehak-ro, Yuseong-gu, Daejeon 34141, Republic of Korea. [2]Green Carbon Research Center, Chemical Process Division, Korea Research Institute of Chemical Technology, 141 Gajeong-ro, Yuseong-gu, Daejeon 34114, Republic of Korea. [3]KAIST Institute for NanoCentury, KAIST, 291 Daehak-ro, Yuseong-gu, Daejeon 34141, Republic of Korea. [4]These authors contributed equally: Jihoon Choi, Keonwoo Choi. ✉e-mail: sgim@kaist.ac.kr; dongyeunkoh@kaist.ac.kr

achieved with a molecular sieve or dense polymeric membranes. However, solute–solute separation within MWCO between 150 and 300 g mol$^{-1}$ is a far less explored field due to the lack of suitable active membrane material equipped with large enough but highly selective pores in a few nanometer range distributed uniformly therein. Numerous high-value APIs are present in this MWCO range, which could benefit from less energy-intensive membrane processes. Zovirax® (Acyclovir, MW: 225.20 g mol$^{-1}$), a market product of GlaxoSmithKline, is widely used as an antiviral drug for treating the Herpes virus. However, due to relatively low bioavailability of Acyclovir (15–30%), various prodrugs of Acyclovir, such as Valtrex™ (Valacyclovir, MW: 324.34 g mol$^{-1}$), famciclovir (MW: 321.33 g mol$^{-1}$), and penciclovir (MW: 253.26 g mol$^{-1}$) have been designed from the Acyclovir to enhance its medicinal effects[9–11]. To enhance medicinal efficacy, the purity of the product should be high, which requires precise separation of these APIs after the synthesis. Unfortunately, only a few membrane processes have succeeded in high-precision separation within the MWCO range of 150–300 g mol$^{-1}$ due to the structural relaxation of polymer chains in the solvent[12,13]. This highlights the need to develop a new membrane system with high selectivity for precise solute–solute separations.

As mentioned previously, constructing an OSN membrane system with chemical stability in diverse harsh solvents and molecular sieving properties has difficulties in both material selection and the membrane fabrication. The membrane system based on inorganic materials with a rigid pore structure continues to face difficulties achieving thin and defect-free film fabrication[14]. Conversely, polymer membrane systems, while being easy to fabricate, have faced limitations in achieving high selectivity due to their relatively large inter-chain free volume. Furthermore, while interfacial polymerization, one of the solution-based membrane fabrication methods, can allow the formation of very thin polyamide films (<100 nm), the membrane morphologies, including thickness and roughness, could not be controlled systematically. Consequently, a crumpled structure was formed that could potentially lead to fouling effects[15]. Therefore, several chemical vapor deposition (CVD) methods have been introduced to precisely control very thin (<100 nm) films. For example, the plasma-enhanced chemical vapor deposition (PECVD) process enables the surface modification of polymeric membranes using low-temperature plasma[16,17]. Successful fabrication of OSN membranes using various monomers, such as acetylene, improved solute rejection and maintained solvent stability. Nonetheless, the complexity of the PECVD mechanism leads to the production of polymer films with partially retained functional groups, thereby restricting their applicability for precision separation[18].

Herein, we suggest an ultrathin, highly interconnected polymer membrane designed for effectively separating small molecules under harsh operating conditions in OSN. To satisfy the desirables, the membrane should possess the cross-linking framework for its mechanical, chemical, and long-term stability, as well as minimal thickness to enhance permeance. One of the advanced CVD technologies, initiated chemical vapor deposition (iCVD) process, offers a thin selective layer at a low substrate temperature (20–50 °C), unlike conventional CVD methods that require high temperatures[19–21]. Since the polymer layer is prepared through surface growth polymerization (initiator decomposition → monomer and radical adsorption → radical polymerization) without any side reactions, it could enhance its integrity with the subjacent, along with allowing for the uniform deposition of a nanoscale film down to 10 nm (Supplementary Fig. 1a)[22]. Furthermore, the iCVD process enables the synthesis of high-purity polymer films without solvents or post-processing, making it feasible to produce defect-free membrane layers. Thanks to the homogeneous mixing of reactants in the vapor phase, it could lead to the facile fabrication of a highly interconnected polymer network[23,24]. Utilizing these advantages, we present the organosiloxane polymer nanolayer with four cross-linkable functionalities, poly(1,3,5,7-tetravinyl-1,3,5,7-tetramethylcyclotetrasiloxane) (pV4D4), as the selective layer, primarily targeting a lower MWCO range of 150–300 g mol$^{-1}$ (Fig. 1a). Within the pV4D4, the organosiloxane bond, –Si–O–Si–, offers flexibility with low bond rotational energy of 0.2 kcal mol$^{-1}$, and the highly interconnected structure, derived from its multifunctionality, enhances mechanical robustness, chemical resilience, and structural longevity in the harsh solvent environment[25,26]. Additionally, flexible octagonal rings and a dense bridging microstructure can play a crucial role in discriminating small molecules at the nanoscale, enabling precise separation. Utilizing porous supports made from cross-linked polyacrylonitrile (XP) membrane, we successfully fabricated an ultrathin, highly interconnected pV4D4 membrane, followed by a series of solvent activation processes to remove unreacted monomers and oligomers within the support membrane (Fig. 1b, c). The prepared pV4D4 membranes were termed based on the measured membrane thickness (in nanometers), the type of support membrane (XP6 or XP12, numeric values represent cross-linking time), and the dimethylformamide (DMF) activation. Notably, we found that the DMF activation of the pV4D4/XP membrane for achieving improved permeance is more effective than other OSN membranes[27]. As a result, the OSN performance after DMF activation could be adjusted with the thickness of the pV4D4 layer and various XP membranes. Furthermore, the 29 nm/XP12-D membranes demonstrated exceptional solute–solute selectivity in the MW range of 150–250 g mol$^{-1}$ and 250–350 g mol$^{-1}$, rendering them more efficient for the energy-intensive, high-value API industry.

## Results
### Morphology of membranes
To support an iCVD nanolayer film, mesoporous polyacrylonitrile (PAN) membranes were prepared using a nonsolvent-induced phase separation[6]. These PAN membranes underwent a series of chemical reactions with a hydrazine solution to enhance chemical stability, resulting in the cross-linked PAN (XP) membranes (Supplementary Fig. 1b–d)[28]. Observations indicated that C≡N groups of PAN converted into cyclic nitrogen-containing structures through cross-linking or formed hydrazide structures with N–H and C=NH groups due to branching (Supplementary Figs. 2a and 3). Depending on the degree of cross-linking, their surface porosity and pore size distribution could be adjusted, resulting in the fabrication of XP6 and XP12 membranes with cross-linking times of 6 and 12 h, respectively. Pore sizes of PAN and XP membranes, measured using gas–liquid porometry, fell within the mesoporous range of 20 nm or less, displaying a narrow pore size distribution. During the cross-linking process, the mean pore size of the XP6 membrane decreased from 17.7 to 14.8 nm (Supplementary Fig. 2b). Unfortunately, monitoring was impossible for XP membranes with cross-linking durations exceeding 12 h due to equipment limitations. However, it can be inferred that pore size and porosity decrease as the cross-linking time increases, as indicated by the decline in pure methanol permeation results (Supplementary Fig. 2c). The morphology of XP membranes was observed using a cross-section scanning electron microscope (SEM). XP6 and XP12 membranes exhibited an asymmetric vertical structure with a denser top layer and a sponge-like porous interior (Fig. 1d, g)[29]. Furthermore, with increasing cross-linking time, surface porosity slightly reduced (Supplementary Fig. 4), and it was observed that the porous skin layer became thicker (Fig. 1d, g and Supplementary Fig. 5).

For the well-controlled coating of pV4D4 on porous XP membranes, we aimed to investigate the process parameters in the iCVD process and the resulting morphology changes of pV4D4/XP membranes. In porous or high-aspect-ratio structures, the iCVD process enables precise control over surface adsorption and diffusion of vaporized reactants, providing overall conformal coating[30]. The surface reaction of the vapor-phase polymerization proceeds with general radical polymerization steps, and the rate of polymerization is

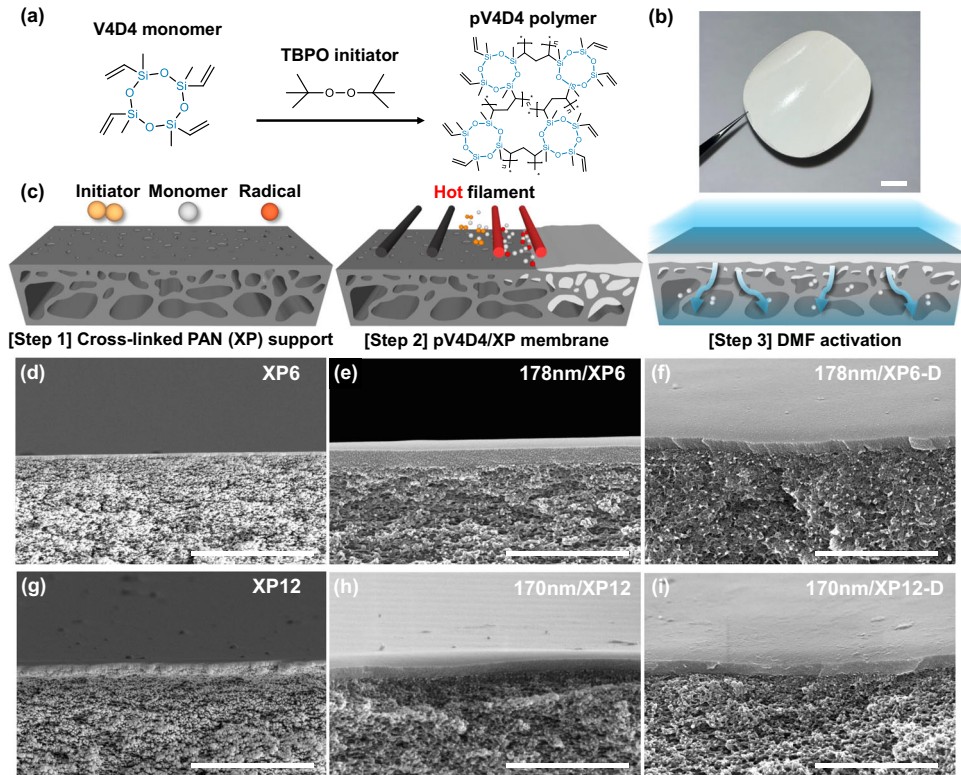

**Fig. 1 | Preparation of pV4D4/XP membranes and cross-section morphology.**
**a** The chemical reaction scheme for the pV4D4 polymer via iCVD process. TBPO represents *tert*-butyl peroxide. **b** Optical image of pV4D4/XP membrane.
**c** Schematic description of pV4D4/XP membrane fabrication and the following DMF activation. **d**–**i** Cross-section SEM images of XP6, 178 nm/XP6, 178 nm/XP6-D, XP12, 170 nm/XP12, 170 nm/XP12-D membranes. Scale bar: 1 cm (**b**), 5 μm (**d**, **g**), 2 μm (**e**, **f**, **h**, **i**).

conditioned by the surface concentration of monomers, i.e., the ratio of partial pressure to the saturated vapor pressure ($P_m/P_{sat}$) in the iCVD process[31,32]. For our purpose of ultrathin, highly smooth membrane coatings, surface polymerization should precede the diffusion of vaporized monomers and radicals into the porous substrate. Moreover, a higher $P_m/P_{sat}$ value of the monomer is required to prevent pore clogging inside the support membrane. The critical factor for the conformality of pV4D4 coatings is the $P_m/P_{sat}$ value of the monomer, which could vary depending on process parameters, including substrate temperature and process pressure associated with surface monomer concentration[33]. After the initial blocking of porous structure mouths, owing to its surface growth polymerization process, the vapor-phase iCVD technique would allow the formation of a nanoscale membrane layer on the top of the mesoporous support surface[30]. In all experimental campaigns, we maintained a high $P_m/P_{sat}$ value of 0.093 for the rate of polymerization to surpass monomer adsorption into the pores prior to excessive condensation of reactants. The top-view SEM images of pV4D4/XP membranes showed a smooth top surface, in contrast to the porous surface of XP membranes (Supplementary Figs. 4b, c and 6a,c). Furthermore, the pV4D4 layers were mostly confined to the top surface of the porous XP membrane (Fig. 1e, h).

However, V4D4 monomers initially diffuse into the surface pores since the absorbed monomers are freely delivered to the support membrane in a vapor state, resulting in the formation of partially percolated inner pores that might increase the resistance of mass transport. The cross-sectional EDS mapping images of the 29 nm/XP12 membrane clearly illustrated the presence of the V4D4-percolated region (Supplementary Fig. 7). As a facile solution, the unreacted monomers and oligomers clogging pores could be removed efficiently by an additional solvent cleaning, where DMF was permeated through the pV4D4/XP membrane for 24 h to eliminate unreacted adsorbents filled in the XP membrane. Following this DMF activation, the pV4D4

membrane formed an intact contact with the porous XP membrane without delamination or structural damage (Fig. 1f, i and Supplementary Fig. 6b, d).

## Depth profile analysis of pV4D4/XP membranes

Time-of-flight secondary ion mass spectrometry (ToF-SIMS) in-depth analysis was performed to investigate the compositional variation at the interface of the pV4D4/XP membranes and their structural changes after the DMF activation step. In the ToF-SIMS spectra, $SiCH_3^+$ and $C_3H_4N^+$ were chosen to trace the structural composition of the pV4D4 layer and XP membrane, respectively (Supplementary Fig. 8a–c). Based on the change in the slope of their ion intensity in the ToF-SIMS spectra, three distinct structural regions were identified (Fig. 2a, b): (1) pV4D4 layer region (Zone I), (2) V4D4-percolated region in the XP membrane (Zone II), and (3) innate XP membrane (Zone III). The layer thickness for each constitutional region was estimated by the product of the calculated etch rate and sputter time (Supplementary Figs. 9, 10 and Supplementary Table 1). Moreover, the resulting pV4D4 thickness closely matched an order of magnitude similar to the pV4D4 layer in SEM images (Fig. 1e, h). In the pristine 29 nm/XP12 membrane, $SiCH_3^+$ ion intensity remained in the sub-micrometer-thick region or above without reaching an XP baseline plateau, indicating the presence of a wide percolation region in the XP membrane (Fig. 2a). After DMF activation, as clearly seen in the ToF-SIMS depth profile spectra, the 29 nm/XP12-D membrane exhibited a substantial reduction in the percolated region. It should be noted that the DMF activation step successfully eliminated unreacted V4D4 monomers and oligomers (Fig. 2b). Compared to the 29 nm/XP12 membrane, the normalized ion intensity of $SiCH_3^+$ dropped to a negligible value during 25–50 s of sputter time for the 29 nm/XP12-D membrane. This occurred at the boundary between the pV4D4 layer and the XP12 membrane (Fig. 2c). It suggests that DMF activation could effectively clear clogged pores in

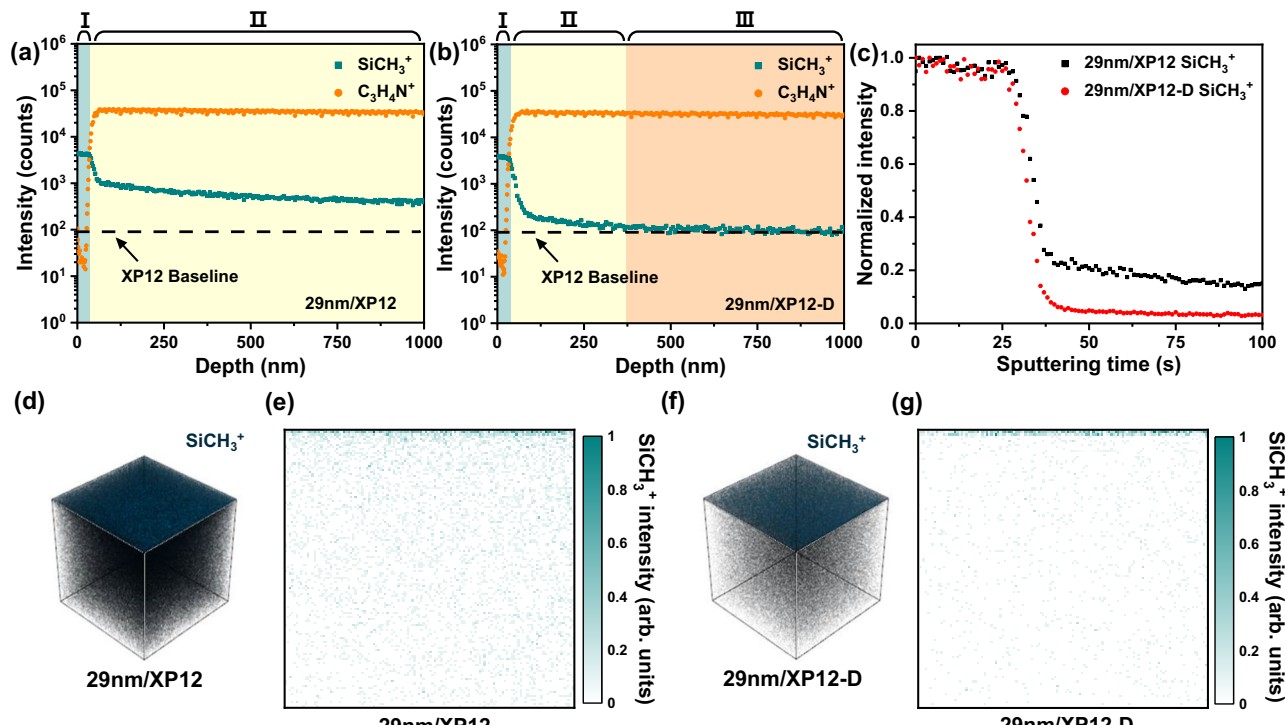

**Fig. 2 | ToF-SIMS depth profile spectra and 3D tomography.** ToF-SIMS depth profile analysis of **a** 29 nm/XP12 and **b** 29 nm/XP12-D membranes. Color: Zone I, cyan; Zone II, yellow; Zone III, orange. **c** Comparison of normalized SiCH₃⁺ ion intensity on the effect of DMF activation. ToF-SIMS 3D tomography and 2D cross-sectional images of (**d**, **e**) 29 nm/XP12 and (**f**, **g**) 29 nm/XP12-D membranes. 3D tomography size: 100 μm × 100 μm × 10 μm.

the V4D4-percolated region, reducing transport resistance and facilitating solvent permeation. In the 3D topographic images of the 29 nm/XP12 and 29 nm/XP12-D membranes, the SiCH₃⁺ ion intensity in the outermost pV4D4 layer region showed no perceptible change. However, it was noticeable that the ion intensity in the region corresponding to the XP membrane decreased significantly (Fig. 2d, f). Additionally, the cross-section results were visually represented in these 2D images, which displayed an ultrathin 29 nm-thick pV4D4 layer on the top of the XP membrane surface with varying SiCH₃⁺ ion intensity distribution according to the DMF activation step (Fig. 2e, g and Supplementary Fig. 8d–g). The boundary between the pV4D4 layer region and the V4D4-percolated region was more clearly distinguished in pV4D4/XP membranes with a thicker pV4D4 layer. Significantly, when the support membrane was changed to the XP6 membrane with a larger pore size, an enlargement of the V4D4-percolated region was also clearly observed (Supplementary Fig. 11). After the DMF activation, the thickness of pV4D4 was slightly reduced from 29 to 28 nm for 29 nm/XP12 (Supplementary Fig. 12). Nevertheless, the vertically deep-formed V4D4-percolated region induces strong adhesion between the pV4D4 layer and the XP membrane, ensuring exceptional stability against structural damage and flux improvement through DMF activation, making the pV4D4/XP membrane highly valuable in high-pressure separation processes.

## Characterizations of pV4D4

The chemical composition of the pV4D4 layer and XP membrane, as well as their integration, was investigated by X-ray photoelectron spectroscopy (XPS) and Fourier transform infrared spectroscopy (FT-IR) analyses. In the XPS spectra, the XP12 membrane exhibited C and N elements from the framework, while the pV4D4/XP12 membrane showed C, O, and Si elements of the organosiloxane structure without an N element from the XP membrane, indicating the successful fabrication of the defect-free pV4D4 layer onto the XP membrane (Fig. 3a). Similarly, the elemental content and high-resolution XPS spectra of the

pV4D4 indicated the presence of a siloxane ring in pV4D4 (Supplementary Fig. 13 and Supplementary Table 2)[34]. In the FT-IR spectra, we observed Si−CH₃ bending at 1261 cm⁻¹, Si−O−Si bending peaks ranging from 1000 to 1130 cm⁻¹, and Si−C rocking between 749 and 791 cm⁻¹ in all membranes except XP12 (Fig. 3b). Furthermore, in the FT-IR spectra of the pV4D4/XP12 membrane, we also identified peaks, including C≡N, that can be attributed to XP12. This observation is likely due to the presence of the thin pV4D4 layer. Additionally, owing to the successful pV4D4 polymerization, the intensity of the C=C stretching peak at 1598 cm⁻¹ slightly decreased compared to that of the V4D4 monomer (Fig. 3c). After V4D4 polymerization, the root-mean-square (RMS) roughness ($R_q$) of the membrane, as determined by atomic force microscopy (AFM), decreased from 1.67 to 1.37 nm (Fig. 3d, e). A pV4D4 layer with reduced roughness could help prevent solute molecule adhesion, potentially alleviating fouling issues[35]. Moreover, the surface roughness of XP12 and 29 nm/XP12 membrane remained consistently low, measuring 3.65 and 1.81 nm, respectively, even after being immersed in DMF for 24 h (Supplementary Fig. 14). To investigate the cross-linking structure of pV4D4, various monomer-to-initiator flow ratios were adjusted. When checking the peak intensity variation representing C=C vibration in FT-IR spectra (1598 cm⁻¹), lower monomer-to-initiator ratios resulted in the lower peak intensity, indicating that a considerable amount of vinyl group participated in the vapor-phase free radical polymerization step, and thereby strongly inferring that the resultant pV4D4 is highly interconnected (Supplementary Fig. 15a, b). Additionally, the refractive index of the pV4D4 films showed an increasing trend as the monomer-to-initiator flow ratios decreased (Supplementary Fig. 15c). The calculated refractive index of pV4D4 films ($n = 1.49$-1.50) was higher than that of other typical organosiloxane polymer, poly(dimethylsiloxane) (PDMS) ($n = 1.41$-1.43), indicating a higher cross-linking density[36–40]. The density of pV4D4 was calculated to be 1.70 g cm⁻³ from the critical angle in X-ray reflectometry spectra, which closely matched the simulation curve (Supplementary Fig. 16). When compared to the

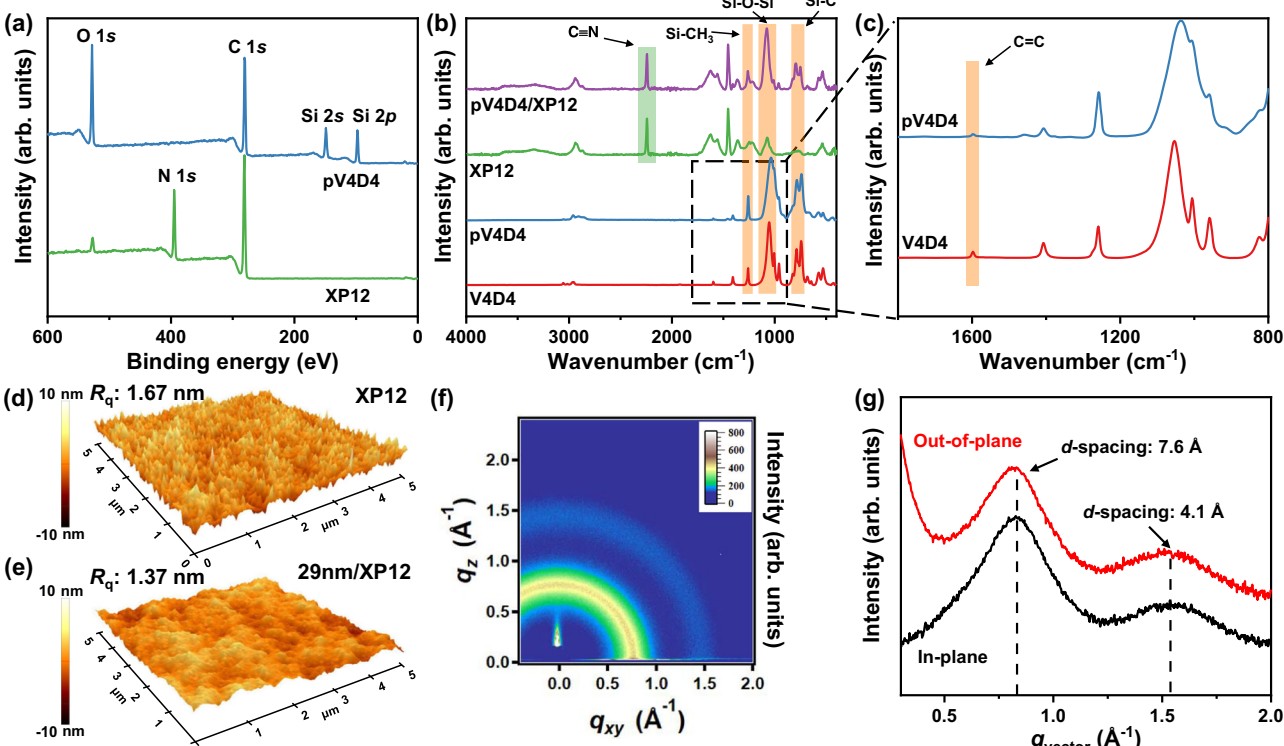

**Fig. 3 | Characterization of pV4D4 polymer. a** XPS survey spectra of pV4D4 polymer on Si wafer and XP12 support membrane. **b** FT-IR spectra of 170 nm/XP12, XP12 membranes, pV4D4 polymer, and V4D4 monomer. **c** The enlarged FT-IR spectra of C=C group absorption band at 1598 cm⁻¹. **d, e** AFM images of XP12 and 29 nm/XP12 membrane. **f** 2D-GIWAXS pattern and **g** 1D-GIWAXS spectra along out-of-plane and in-plane axis of pV4D4 polymer film.

densities of other silicone polymers, such as PDMS (0.97 g cm⁻³) or 1,2-bis(triethoxysilyl)ethane (1.50 g cm⁻³), this high-density value indicates the formation of a more highly interconnected pV4D4 network than other silicone polymers (Supplementary Table 3)[41,42]. Notwithstanding the highly interconnected nature of the ultrathin membrane layer, the pV4D4/XP membrane shows no noticeable compromise in mechanical flexibility (Supplementary Fig. 17).

The structural property of pV4D4 was investigated using grazing-incidence wide-angle X-ray scattering (GIWAXS). The out-of-plane ($q_z$, $q_{out}$) scattering vector reflects the face-on orientation of the pV4D4 chains constituting the intrinsic transport channels, which corresponds to $\pi$−$\pi$ stacking. Conversely, the in-plane ($q_{xy}$, $q_{in}$) scattering vector reflects the edge-on packing perpendicular to the substrate, corresponding to lamellar stacking. Both scattering vector intensities exhibit amorphous halo diffraction in the range of 0.5–1.0 Å⁻¹ and around 1.5 Å⁻¹, indicating the presence of intrinsic pore channels within the highly interconnected network (Fig. 3f). Correspondingly, the vertical line-cut with respect to $q_z$ reveals two peaks indicating inherent transport channels of ~4.1 and 7.6 Å $d$-spacing within the pV4D4 layer (Fig. 3g). These two $d$-spacing values are speculated to originate from the unique structure of the pV4D4 network. The Si–O bonds within the octagonal siloxane ring create strong through-space conjugation, while the alkyl chain promotes the self-organization of organic molecules[43,44]. Therefore, $d$-spacing of 4.1 and 7.6 Å might be attributed to the siloxane ring and alkyl chain, respectively, indicating the capability for efficient solute–solute OSN (MWCO 150–300 g mol⁻¹).

**Organic solvent nanofiltration**

Organosiloxane membranes were initially tested for single gas permeation to check the integrity and intrinsic transport properties of the pV4D4 layer as a membrane. For gas permeation tests, pV4D4/XP membranes were prepared to involve various pV4D4 layer thicknesses

and XP membranes with different cross-linking times. The He/N₂ ideal selectivity for the pV4D4 layer was used as an indicator for checking the presence of large pinhole defects in the membrane. The He/N₂ ideal selectivity for pV4D4/XP membranes was summarized in Supplementary Table 4. The 7 nm/XP6 showed a He/N₂ ideal selectivity of 2.69, which corresponds to the Knudsen selectivity of the non-selective transport pathway, resulting from insufficient coverage of the pV4D4 layer over the XP6 membrane. As the iCVD deposition time was increased, still relatively short, within 30 min, the 16 nm/XP6 exhibited an enhanced He/N₂ ideal selectivity of 9.10, indicating the elimination of nanoscopic pinhole defects in the selective layer. The He/N₂ ideal selectivity was further improved to 17.88 for 55 nm/XP6 with a thicker pV4D4 layer. By utilizing different probe gas molecules (He, CO₂, O₂, N₂, and CH₄), the overall gas permeance of the 55 nm/XP6 decreased as the kinetic diameter of each gas increased, suggesting the molecular sieving property of the fabricated pV4D4/XP membranes (Supplementary Fig. 18). Moreover, the He/N₂ ideal selectivity observed in the 35 nm/XP12 membrane after DMF activation decreased from 12.35 to 6.38, and the concomitant increase in gas permeance is speculated to be associated with the reduction in the V4D4-percolated region.

To balance the membrane integrity and membrane's productivity (i.e., the permeance of molecules through the membrane layer, higher when membrane thickness is lower), we prepared membranes with pV4D4 layer thickness in the range of 29−35 nm on two different types of support membranes of XP6 and XP12. To fully understand the intrinsic solvent flux through the membrane, we first proceeded with organic solvent permeation tests with 29 nm/XP12 by changing solvents (methanol → acetone → methanol → DMF → methanol) over a 5-day period in a custom-built cross-flow setup (Fig. 4a and Supplementary Fig. 19). In the initial 24 h of methanol permeation, the 29 nm/XP12 membrane exhibited a low methanol permeance of 0.042 L m⁻² h⁻¹ bar⁻¹, which gradually increased over time (Fig. 4a). After 24 h, the permeating solvent was changed to acetone, and the

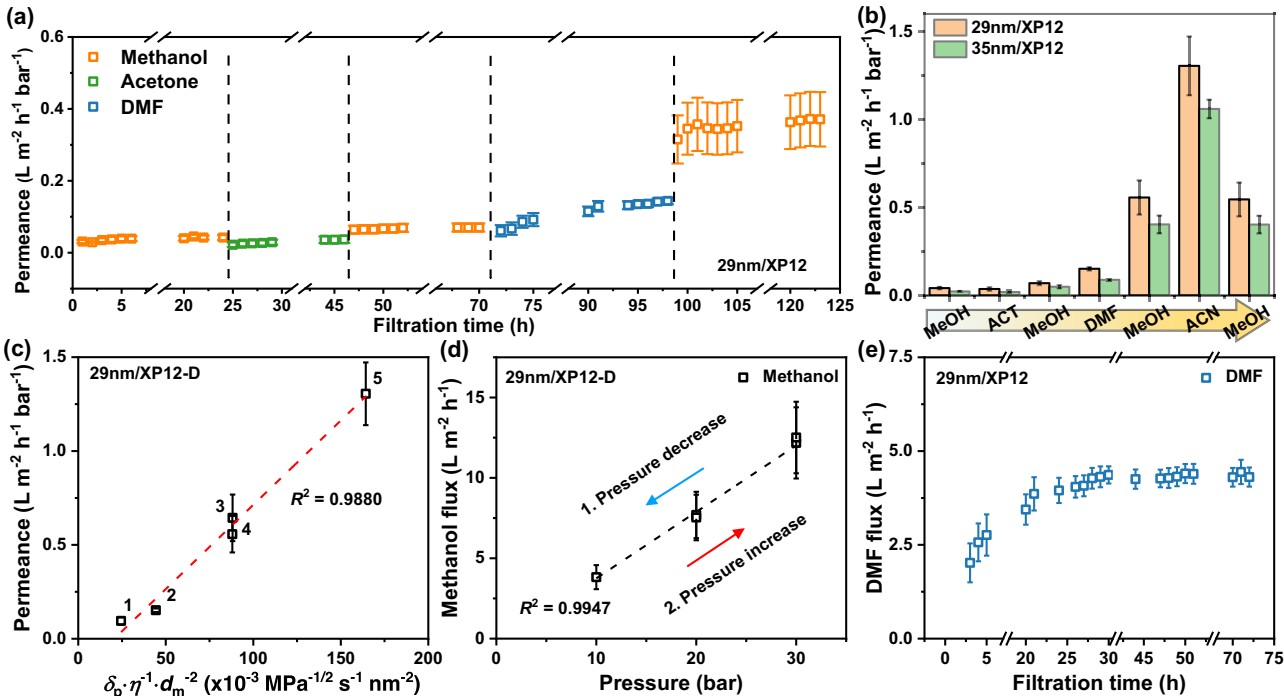

**Fig. 4 | Solvent permeation performances. a** Solvent permeation test of 29 nm/XP12 membrane with time and changing the solvents (methanol, acetone, and DMF) under 30 bar pressure in a custom-built cross-flow system. **b** Change of methanol permeance of 29 nm/XP12 and 35 nm/XP12 membranes with various activation solvent. MeOH, ACT, and ACN represent methanol, acetone, and acetonitrile, respectively. **c** Various solvent permeances of 29 nm/XP12-D membrane was correlated with the combined solvent property ($\delta_p \cdot \eta^{-1} \cdot d_m^{-2}$). 1: ethanol, 2: DMF, 3: acetone, 4: methanol, 5: acetonitrile. **d** Methanol flux of 29 nm/XP12-D membrane with varying pressures. **e** DMF flux of 29 nm/XP12 membrane with filtration time. All error bars indicate the standard deviation at two or three measurements.

permeance through the same membrane also showed a slight increase over time. Then, when the solvent was changed to methanol again, a slightly enhanced methanol permeance of 0.070 L m⁻² h⁻¹ bar⁻¹ was observed for another 24 h. As the fourth sequential solvent, DMF was applied to the membrane for permeation. DMF permeance through the membrane was gradually increased over time until the plateau for permeation reached around 30 h of DMF permeation. Finally, the membrane was subjected to methanol permeation again, and a tenfold increased methanol permeance of 0.42 L m⁻² h⁻¹ bar⁻¹ was observed. The enhanced permeance effect by the DMF treatment, compared to other solvents, aligned well with the depth analysis result that the DMF activation step would remove unreacted monomers and oligomers in the V4D4-percolated region, as observed by ToF-SIMS depth profiles. When the membrane thickness was increased to 35 nm, the overall permeances decreased for the similar sequential solvent permeation tests (Fig. 4b); however, the DMF activation was still effective in achieving the intrinsic permeance through the pV4D4/XP membrane. Organosiloxane membrane showed remarkable stability under various harsh solvents, including DMF, showing the wide range of organic solvent separation could be demonstrated by the pV4D4 membrane.

After the DMF activation, the permeance of different organic solvents with 29 nm/XP12-D membrane was investigated. Supplementary Table 5 displays the physicochemical parameters of the chosen solvents. From these observations, the combined solvent property ($\delta_p \cdot \eta^{-1} \cdot d_m^{-2}$), which includes the Hansen solubility parameter for polar ($\delta_p$), viscosity ($\eta$), and molar diameter ($d_m$), was shown to be linearly related to solvent permeance (Fig. 4c). Poor correlation was observed between the solvent permeance and inverse of viscosity ($\eta^{-1}$) or other combined solvent properties with different solubility parameters related to dispersion ($\delta_d$), hydrogen bond ($\delta_h$), and total ($\delta_t$) (Supplementary Fig. 20)[15]. Further to this, the methanol flux of the activated 29 nm/XP12-D was evaluated to be 11.1 L m⁻² h⁻¹ regardless of time. A

linear relationship between applied pressure and methanol flux was found, indicating that the operating conditions are stable in the pressure range of 10–30 bar (Fig. 4d and Supplementary Fig. 21). The DMF activation process was maintained for ~24 h, followed by a constant DMF flux reaching a steady state after 3 days, exhibiting chemical stability of the pV4D4 membrane without its structural degradation (Fig. 4e). Resistance to time, pressure, and harsh solvents was speculated to be conferred by the unique chemical structure of the highly interconnected pV4D4, which possesses low free volume.

To understand the solute selectivity of the organosiloxane membrane, polystyrene (PS) rejection profiles in various solvents were determined. To investigate how the PS rejection profiles of the pV4D4 membranes vary with different support membranes with different pore sizes, 30 nm/XP6-D and 29 nm/XP12-D membranes were prepared. For the 30 nm/XP6-D membrane, broad PS rejection profiles with high MWCO > 1000 g mol⁻¹ were found in all solvents (Fig. 5a). In contrast, for the 29 nm/XP12-D membrane, PS rejection profiles, which were the MWCO of 252 g mol⁻¹ for methanol, 255 g mol⁻¹ for acetone, and 369 g mol⁻¹ for DMF, were estimated (Fig. 5b). These large differences in PS rejection profiles highlighted the importance of smaller surface pores, which reduce surface roughness and the diffusive penetration of monomer vapors into the substrate during the iCVD process. Meanwhile, the PS rejection of 29 nm/XP12-D membrane with a MW < 370 g mol⁻¹ slightly decreased in DMF solvent permeation. Nevertheless, the PS rejection profile remained almost unchanged across different filtration times (~24 h), indicating the stability of the pV4D4/XP membrane (Supplementary Fig. 22). As the thickness of the pV4D4 layer increased to 35 nm, a remarkable enhancement in the rejection of solutes was obtained due to the formation of a tighter structure (Fig. 5c). Supplementary Table 6 lists all PS rejection data. Intriguingly, the sharp PS rejection curve for 29 nm/XP12-D membranes between 162 and 266 g mol⁻¹ MW in methanol and acetone

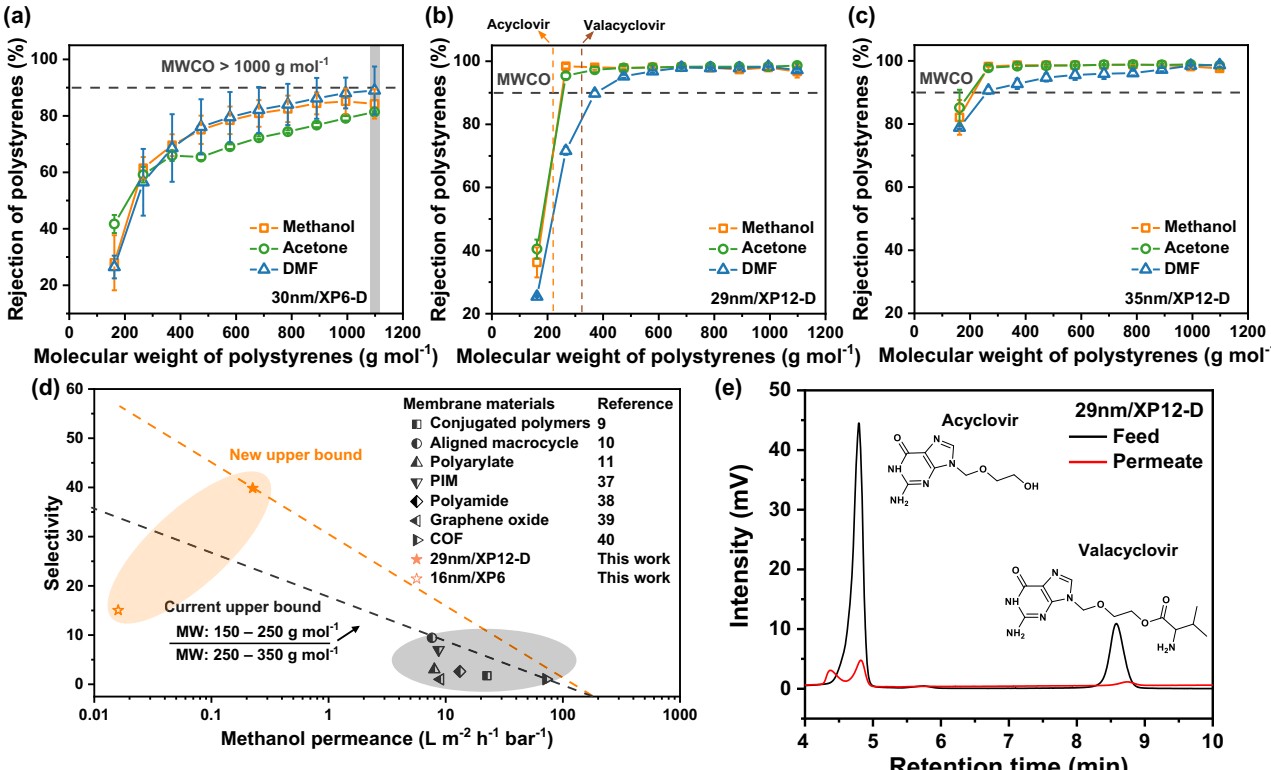

**Fig. 5 | Polystyrene rejection profiles, comparison with other state-of-the-art membranes, and high-performance liquid chromatography (HPLC) analysis of API products (Acyclovir and Valacyclovir).** The rejection profiles of **a** 30 nm/XP6-D, **b** 29 nm/XP12-D, and **c** 35 nm/XP12-D membrane with different molecular weights of polystyrene oligomers in methanol, acetone, and DMF and their molecular weight cut-off (MWCO) for each solvent. **d** Trade-off relationship between methanol permeance and solute–solute selectivity and new upper bound with the MW range of 150–250 g mol⁻¹ over 250–350 g mol⁻¹. PIM and COF represent polymer of intrinsic microporosity and covalent organic framework, respectively. **e** Ultra-violet absorbance of API products (Acyclovir and Valacyclovir) was measured by HPLC at both and permeates of 29 nm/XP12-D, corresponding to retention time. All error bars indicate the standard deviation at two or three measurements.

solvents suggests that solutes with MW: 150–250 g mol⁻¹ and MW: 250–350 g mol⁻¹ can be effectively separated from each other. The 29 nm/XP12-D membranes demonstrated superior selectivity in the MW: 150–250 g mol⁻¹ and MW: 250–350 g mol⁻¹ ranges when compared to membranes reported in the literature (Fig. 5d and Supplementary Table 7)[12–14,45–48]. By modifying the surface pore size of the support membrane and activating the membrane with DMF, the pV4D4/XP membranes achieved improved permeance and solute selectivity. This enhancement was particularly evident for solutes with MW: 150–250 g mol⁻¹ and 250–350 g mol⁻¹, surpassing the current upper bound. To demonstrate the effectiveness of the 29 nm/XP12-D membrane in purifying API products, we further evaluated the separation of Acyclovir (MW: 225.20 g mol⁻¹) and Valacyclovir (MW: 324.34 g mol⁻¹) in methanol. This solute pair (i.e., solute–solute) matches the MWCO range provided by 29 nm/XP12-D that solute molecules in 150–250 g mol⁻¹ (i.e., Acyclovir) can be separated from the solute molecules in 250–350 g mol⁻¹ (i.e., Valacyclovir). The permeation of Acyclovir and Valacyclovir through 29 nm/XP12-D was tested with methanol, and HPLC results showed retention times of 4.8 and 8.6 min for Acyclovir and Valacyclovir, respectively (Fig. 5e). The split peak observed in the permeate at 4.4 min was due to the formation of a potential intermediate of Acyclovir in the HPLC solvents[49]. The calculated rejections were 88.95 ± 3.26% and 99.00 ± 0.53% for Acyclovir and Valacyclovir, respectively. Despite a MW difference of <100 g mol⁻¹ between Acyclovir and Valacyclovir, the high solute–solute selectivity between these solutes was achieved as 11.04, providing the precision API separation that has previously been unattainable in polymer-based membranes. In energy intensive, high-value API industries, precise separation of solutes using a membrane with high selectivity offers a

challenge that is difficult to overcome even with an ideal isotropic membrane[14,50,51]. Our work will be extended to industries requiring precise molecular separation by offering materials and fabrication technologies adequate for the development of large-scale, thin, defect-free membranes important for membrane science.

## Discussion

We present a strategy for fabricating an ultrathin, highly interconnected all-polymer membrane using the iCVD process. The highly interconnected network formed from organosiloxane-class pV4D4 imparts solute–solute selectivity as well as satisfactory durability in OSN operation. With precise control enabled by the iCVD process, it allows for the deposition of an ultrathin pV4D4 layer down to 16 nm on the XP membrane. The pore size of the support membrane is optimized for integrating the pV4D4 selective nanolayer. Corresponding membranes undergo DMF activation, removing unreactants for improved permeance, ensuring their sufficient pores without structural compaction. The 29 nm/XP12 membrane showed extraordinary solute–solute selectivity in the unexplored MW range of 150–300 g mol⁻¹, surpassing that of most state-of-the-art OSN membranes.

Based on our findings, membranes demonstrating stable permeance and selectivity over extended periods under varying pressure and various solvent conditions hold great potential for OSN applications. Highly interconnected organosiloxane materials face challenges in membrane fabrication due to high viscosity, gelation behavior properties in bulk, and limitations in available solvents. Our successful fabrication of organosiloxane membranes using the solvent-free iCVD method, capable of synthesizing defect-free and high-quality highly

interconnected polymer films without undesired reactions, not only presents previously inaccessible materials but also suggests a powerful approach for fabricating highly selective and physicochemically stable membranes. With this emerging process, copolymerization using a wide range of monomers given in the iCVD libraries will facilitate the elaborate design of next-generation high-performance iCVD membranes[52,53].

## Methods

### Synthesis of support membranes

Mesoporous PAN support membranes were prepared by the nonsolvent-induced phase separation method with a viscous polymer solution of PAN:1,3-dioxolane:DMSO at a w/w ratio of 10:45:45[6]. A transparent dope solution was formed on a jar roller at 75 °C to make a homogeneous solution, followed by manual degassing in a syringe pump (Teledyne Isco, 500D). The polymer solution was filtrated through a 15 μm stainless steel in-line filter at up to 15 bar pressures. The PAN support membranes were cast on a polypropylene/polyethylene non-woven fabric (Novatexx 2471, Freudenberg Filtration Technologies, Germany) using an automatic casting machine with a casting speed of 2.5 m/min, a casting thickness of 200 μm and a web width of 0.3 m. Immediately, the membrane was immersed into a deionized water bath at room temperature and transferred in a hot deionized water bath at 80 °C for 3 h to remove excess solvent[54,55]. After cooling, the membranes were rinsed with deionized water for 1 day and immersed in an IPA bath. Upon drying the PAN support membranes at room temperature, they were placed in a 20% (v/v) concentration of hydrazine monohydrate in deionized water at 85 °C for 6 or 12 h[28,56,57]. The resultant cross-linked PAN (XP6 or XP12) membranes were rinsed with deionized water for 1 day, stored in an IPA bath, and dried in a vacuum oven at room temperature before use. For membrane characterization, membranes cast without a non-woven fabric were used.

### Synthesis of iCVD pV4D4 membranes

V4D4 and *tert*-butyl peroxide (TBPO) were vaporized and delivered to a custom-built iCVD reactor. The flow rates of V4D4 and TBPO were 1.61 and 1.62 sccm, respectively, controlled by a needle valve. For the fabrication of the pV4D4 membrane, the iCVD process conditions were as follows: the process pressure was 200 mTorr, the filament was heated to 140 °C, and the substrate temperature was maintained at 42 °C. The deposition rate of the pV4D4 layer was 1.41 nm min⁻¹ and the $P_m/P_{sat}$ value was calculated as 0.093.

### Organic solvent nanofiltration (OSN)

The permeance of organic solvents and PS rejection with various MWs were estimated. Most nanofiltration experiments were conducted at room temperature and 30 bars in a custom cross-flow system. The effective membrane area in each cell was 14.2 cm². The cross-flow system consisted of three cells in series with a feed flow of 50 L h⁻¹. A few experiments of support membranes were carried out in a custom dead-end filtration system at room temperature, 2 bars, and the effective membrane area was 15.2 cm². Permeate for permeance measurements were obtained at 1 h intervals, and rejection estimations were measured for permeates collected after 24 h, when the filtration system had reached steady state. Before the solute rejection test, the chosen pure solvent was filtered for 1 day throughout the membrane to eliminate impurities. For the single solvent filtration system, solvents, including MeOH, EtOH, ACT, ACN, and DMF were used. A feed solution consists of PS dissolved in MeOH, ACT, and DMF were used. The PS mixture comprised 0.5 g L⁻¹ of PS 750 (PDI = 1.09; Shimadzu Scientific Korea) and 0.1 g L⁻¹ of PS 162, PS 266, and PS 370 (PDI = 1; Shimadzu Scientific Korea). To ensure distinct HPLC peaks for PSs with a MW of 266 g mol⁻¹ or above, we first evaporated all solvents and subsequently re-dissolved them in ethanol before HPLC analysis[58]. However, for PS with a MW of 162 g mol⁻¹, due to its volatile nature, we analyzed it directly without

any treatment. We investigated PS rejection using a YL9100 HPLC system with an ultraviolet-visible detector set at 254 nm. Using a reverse-phase column (ACE 5-C18-100, 250 × 4.6 mm), HPLC analysis was conducted (Advanced Chromatography Technologies). The mobile phase comprised 35 vol% of HPLC-grade water, 65 vol% of tetrahydrofuran, and 0.1 vol% of trifluoroacetic acid. The analysis of API products (Acyclovir and Valacyclovir) was performed using the same conditions as the PS analysis, with the exception that the mobile phase consisted of 95 vol% water, 5 vol% acetonitrile, and 0.19 vol% formic acid.

The solvent permeance (*P*) was calculated as follows:

$$P = \frac{V}{A \cdot \Delta t \cdot \Delta p} \tag{1}$$

where *V* is the collected permeate volume (L), *A* is the membrane area (m²), $\Delta t$ is the time required to collect the permeate volume (h), and $\Delta p$ is the transmembrane pressure (bar). The industry standard of liters per square meter per hour per bar (L m⁻² h⁻¹ bar⁻¹) was used to determine the permeance. The permeance was analyzed using the transport model proposed for the OSN membrane. In this model, the hydraulic permeance of the organic liquid through the pV4D4 membrane was calculated according to the following equation:

$$P_s = K \cdot \left( \frac{\delta_p}{\eta_s \cdot d_{m,s}^2} \right) \tag{2}$$

where, $\delta_p$ is the Hansen solubility parameter, $\eta_s$ is the solvent viscosity, $d_{m,s}$ is the molar diameter, and K is the proportionality constant.

The PS rejection ($R_i$) was calculated as follows:

$$R_i = \left( 1 - \frac{C_{p,i}}{C_{F,i}} \right) \cdot 100 \, (\%) \tag{3}$$

where $C_{p,i}$ and $C_{F,i}$ represent the concentrations of the styrene oligomers in the permeate and the feed, respectively. Selectivity of MW of 150–250 g mol⁻¹ against 250–350 g mol⁻¹, two styrene oligomers PS 162 (162 g mol⁻¹) and PS 266 (266 g mol⁻¹) were used to calculate the selectivity:

$$\text{Selectivity} = \frac{C_{permeate,small}/C_{feed,small}}{C_{permeate,big}/C_{feed,big}} \tag{4}$$

where $C_{permeate,small}$ and $C_{feed,small}$ represent the concentrations of the small rejection marker in the permeate and the feed, $C_{permeate,big}$ and $C_{feed,big}$ represent the concentrations of the big rejection marker in the permeate and the feed, respectively.

## Data availability

All data supporting the results of this study are available within the article and its supplementary information file. Source data are available from corresponding authors upon request.

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

## Acknowledgements

This research was supported by Basic Science Research Program through the National Research Foundation of Korea (NRF) funded by Ministry of Science, ICT & Future Planning (No. 2021R1C1C1012014) and National R&D Program through the National Research Foundation of Korea (NRF) funded by Ministry of Science and ICT (Grant No. NRF-2022R1A5A1033719). This work was also supported by the National Research Foundation of Korea (NRF) grant funded by the Korea government (MSIT) (No. 2021R1A2B5B03001416) and the Technology Innovation Program (1415181712, RS-2022-00144300) funded by the Ministry of Trade, Industry & Energy (MOTIE, Korea).

## Author contributions

J.C., K.C., S.G.I. and D.-Y.K. conceived the research concept, designed the experiments, and wrote the manuscript. J.C. and K.C. designed and prepared all the support membranes and pV4D4/XP membranes. J.C. conducted all organic solvent nanofiltration tests. K.C. performed the membrane and material characterization. Y.K., D.K. and Y.Y. assisted in the membrane fabrication experiments and solvent permeation tests. All authors participated in the development of the research discussion. S.G.I. and D.-Y.K. supervised the study. All authors approved the final version of the manuscript.

## Competing interests

The authors declare no competing interests.
