## [Peer Review File · Nature Communications]

Ultrathin Organosiloxane Membrane for Precision Organic Solvent NanofiltrationReviewers' Comments:

Reviewer #1:

Remarks to the Author:

This work describes the fabrication and testing of crosslinked organosiloxane organic solvent nanofiltration (OSN) membranes. iCVD is used to deposit a cyclic siloxane polymer onto the surface that allows for ultra-thin selective layers on top of a commercially available membrane support. There is thorough characterization of the material to understand the composition and performance with multiple organic solvents. The high selectivity allows for a potential new upper bound line to be formed for methanol OSN. This work should be published after minor revisions. Questions and comments are listed below.

1. Are there any EDS measurements of the membrane cross section to accompany the many SEM images? Many of the arguments around the siloxane polymer infiltration into the support could be bolstered with EDS mapping of the SEM cross sections.
2. How much material is lost during the DMF washing? What percentage of the deposited siloxane polymer remains attached to the final membrane after processing.
3. Can you comment on the stability of these membranes in the presence of acids and bases? Many organic solvent waste streams from chemical processes are not near neutral pH and that can considerably impact the membrane performance.
4. The permeance of these membranes is quite low even with 30 nm thick selective layers. What strategies could be employed to improve the permeance of these membranes towards viable levels (~ 10 LMH/bar)?
5. The polystyrene solution used for rejection testing was fairly dilute in concentration. Were any other solutes of higher concentration tested with these membranes? Do you maintain high rejection under more concentrated feed conditions?
6. What is the fouling/regeneration behavior of these membranes when they are exposed to higher concentration solutions?
7. Can you comment on the mechanical properties of the membranes before and after iCVD deposition? Does the addition of the highly crosslinked siloxane significantly decrease the membrane flexibility?
8. Do these membranes undergo loss of permeance after aging?

Reviewer #2:

Remarks to the Author:

The manuscript by Dong-Yeun Koh and coworkers reports an ultrathin, cross-linked organosiloxane membrane for precision organic solvent nanofiltration. The fabrication of organosiloxane membrane using CVD method has been reported (J. Membr. Sci. 230 (2004) 49–59). The development of this work is using (1,3,5,7-tetravinyl-1,3,5,7-etrarmethylcyclotetrasiloxane) (pV4D4) as the precursor. However, the advantage of the new precursor is not fully presented. The membrane separation performance is not comparable with the reported polyamide membrane. The manuscript doesn't meet the standards that make it improper to publish in Nature Communications.

1. The author mentioned that the resulting membranes were heavily cross-linked membranes, how do we define them heavily? There is no specific characterization to certify high cross-linking degree of the material in comparison with other polymers, and it is also lack of corresponding experiments to regulate the cross-linking of the membrane. Besides, the author also emphasizes the high-density of the membrane, the relationship between the two concepts should be clearly expounded.
2. The membrane showed the selectivity of the molecule with the MWCO of 150-300 and 250-300 g mol⁻¹, but the significance of molecular sieve fractions at this scale is not clearly illustrated. In addition, IP is a widely investigated process of NF and RO membrane fabrication, the thickness, crosslinking degree and screening performance of the selective layer can be well regulated in previous reports. The IP based membrane with ~ 10 nm thickness and low MWCO of ~ 200 Da with high permeability have been reported in previous studies. In this work, the membrane shows the lower flux

with similar MWCO value, the advantages of the iCVD are not well demonstrated which makes this work less innovative.

3. According to Figure 5, the support substrate will affect the PS rejection, so which functional layer ultimately determines the selectivity of the membrane?
4. Meanwhile, the manuscript mentioned the resulting membrane was an ultrathin membrane, but the methanol permeability of the membrane is very low, and I did not see any highlight performance in the organic solvent nanofiltration.
5. How to tune the pore size as the author mentioned in Abstract? Detailed experimental results are needed?
6. This sentence "the 29nm pV4D4 nanoscale selective layer fabricated in the vapor phase provides a homogeneous, narrow molecular sieving property, enabling a record-high solute selectivity of 39.88 in the range of between 150 – 250 g mol⁻¹ and 250 – 350 g mol⁻¹." is unclear in expression. Why are the two ranges involved?
7. Why the membrane needs to undergo a 24-hour restructuring in DMF?
8. During the iCVD process, the unreacted monomers are easy to deposit on the substrate membrane. In this work, the authors tried to remove the monomers and by-products by DMF treatment. However, during the separation process, the flux of DMF increases with increasing the filtration time which is different from other solvents. Whether it is due to the membrane swelling or incompletely removal of by-products?

Reviewer #3:

Remarks to the Author:

This work reports a highly selective organosiloxane membrane prepared by iCVD for organic solvent nanofiltration (OSN) applications. As the authors highlighted, the precise separation of small molecules with an Mw range of 150–300 g/mol is very important, which has been overlooked by many studies. Overall, I felt the authors could successfully address the main challenge in the OSN field, supported by their novel ideas and solutions. I recommend its publication after considering my several comments to improve this work.

1. The authors claim that the prepared pV4D4 membranes are "heavily" crosslinked, but I could not find the rationale behind the term "heavily". In other words, have the authors estimated the crosslinking density of the pV4D4 membranes? Specifically, I was concerned about the presence of a C=C peak in pV4D4 (Fig. 3c), which indicates that the presence of unreacted monomer is still significant. I understand that the crosslinking density estimation is very challenging for thin films, so I recommend the authors use an alternate expression throughout the paper (for example, just "ultrathin organosiloxane membrane" would be fine in the title).
2. Can the authors explain more details about the density estimation by XRR? For example, is it bulk density or true density?
3. I felt more details about the structure-property-performance relationship are needed. The authors only characterized d-spacing as a potential contributor to the OSN performance. Is there any other way to estimate the structural properties of the pV4D4 materials (e.g., BET?).
4. In Fig. 5, it is interesting to observe the subtle change in membrane thickness (only ~6 nm) between 29nm/XP12-D and 35nm/XP12-D resulting in a huge difference in the rejection profiles. Could the authors comment on the reason behind this behavior?
5. Can the authors provide more details on the polystyrene rejection estimation? For example, some people use solvent transfer to achieve distinguishable HPLC peaks.
6. For GIWAXS analysis, the authors mentioned it was measured using a pV4D4 film coated onto a

silicon substrate. How did the authors prepare the sample? Did the authors transfer the film from the XP substrate to the silicon substrate?

7. Is there any theoretical basis for the proposed new upper bound in Fig. 5d?

8. The represented density of polyimide (1.4 g/cm³) can be significantly varied depending on which type of polyimide (high or low free volume). Please consider it.

9. The designation of sample names is quite confusing. For example, I assume that 178 nm/XP6 means that 178 nm of pV4D4 was coated onto the XP6 membrane, but no definition was provided in the main text. In this sense, what is the sample shown in Fig. S7a?

Reviewer #1

Remarks to the Author:

This work describes the fabrication and testing of cross-linked organosiloxane organic solvent nanofiltration (OSN) membranes. iCVD is used to deposit a cyclic siloxane polymer onto the surface that allows for ultra-thin selective layers on top of a commercially available membrane support. There is thorough characterization of the material to understand the composition and performance with multiple organic solvents. The high selectivity allows for a potential new upper bound line to be formed for methanol OSN. This work should be published after minor revisions. Questions and comments are listed below.

1. Are there any EDS measurements of the membrane cross section to accompany the many SEM images? Many of the arguments around the siloxane polymer infiltration into the support could be bolstered with EDS mapping of the SEM cross sections.

Thank you for your considerate comments. We agree with the Reviewer that an EDS mapping can be highly useful to assess the qualitative distribution of the siloxane polymer infiltration in the support. In alignment with the SEM analysis results in the manuscript, the EDS mapping also clearly exhibits the presence of the siloxane polymer, especially Si and O elements, which are well spread in the cross-sectional direction of the membrane. **Supplementary Figure 7** was modified accordingly by adding the newly obtained EDS results to better understand siloxane polymer infiltration into the support.

The following sentence was also added/modified in the main manuscript.

- **Page 9, Lines 185:** The cross-sectional EDS mapping images of the 29nm/XP12 membrane clearly illustrated the presence of the V4D4-percolated region (**Supplementary Figure 7**).
- **Supplementary Fig. 7.** was added.

Supplementary Fig. 7. Cross-sectional SEM image and elemental mapping images of 29nm/XP12 membrane. (a) Cross-sectional SEM image of the 29nm/XP12 membrane, along with (b-d) elemental mapping images of carbon, oxygen, and silicon, respectively. Scale bar: 3 μm

2. How much material is lost during the DMF washing? What percentage of the deposited siloxane polymer remains attached to the final membrane after processing.

As the Reviewer pointed out, the DMF washing removed the unreacted monomeric and/or oligomeric substances in the pV4D4/XP membrane. In the ToF-SIMS analysis, although the SiCH_3^+ ion intensity in the V4D4-percolated region changed significantly before and after DMF washing (**Figure 2c**), quantifying the amount of material lost during the DMF washing was still extremely challenging since

the total contents of the deposited ultrathin siloxane polymer were too small compared to the whole volume of the support membrane. Instead, to quantitatively determine the variation in the deposited siloxane polymer after DMF washing, we re-processed the ToF-SIMS data with a better resolution to precisely estimate the thickness of the pV4D4 layer before and after DMF washing. The re-processed data on the pV4D4 thickness change of 29nm/XP12 is shown in the newly added **Supplementary Figure 12**. The pV4D4 thickness, determined as the cross point of the SiCH_3^+ and $\text{C}_3\text{H}_4\text{N}^+$ ion intensities, was indeed slightly reduced after the DMF washing. The 29 nm pV4D4 layer decreased to 28 nm, and the corresponding thickness changes were -3.4 %. However, it is worth noting that the loss of the deposited siloxane polymer does not indicate poor membrane stability of structural compaction, which was confirmed by the stable OSN operation during DMF permeation for 72 hours (**Figure 4e**).
The following sentence was added/modified in the main body.

- **Page 11, Lines 227:** After the DMF activation, the thickness of pV4D4 was slightly reduced from 29 nm to 28 nm for 29nm/XP12 (**Supplementary Figure 12**).
- **Supplementary Fig. 12.** was added.

Supplementary Fig. 12. ToF-SIMS depth profile of 29nm/XP12 and 29nm/XP12-D. SiCH_3^+ and $\text{C}_3\text{H}_4\text{N}^+$ ion intensities of 29nm/XP12 and 29nm/XP12-D.

3. Can you comment on the stability of these membranes in the presence of acids and bases? Many organic solvent waste streams from chemical processes are not near neutral pH and that can considerably impact the membrane performance.

As the Reviewer pointed out, many organic solvent waste streams often do not have a neutral pH. Similarly, active pharmaceutical ingredient (API) purification in the pharmaceutical industry usually operates in acidic or alkaline conditions. To demonstrate the stability of the membrane under such non-neutral pH conditions, we additionally conducted a series of OSN experiments applied to methylene blue (MB, MW: 319.85 g mol⁻¹) in methanol solution. The new experiments used two representative pH levels: pH 3 and 11 (**Figure for Review** below). The pH of the solution was adjusted using 0.1 M HCl and 0.1 M NaOH aqueous solutions, and the dye concentration was set to 15 mg L⁻¹. The nanofiltration test was conducted using a dead-end stirred cell (300 rpm) at room temperature and 30 bars, and the MB concentration in the permeate was estimated by ultraviolet-visible (UV-Vis)

Figure for Review. Methylene blue (MW: 319.85 g mol⁻¹) rejection at pH 3, neutral pH and pH 11.

spectroscopy. The OSN test was conducted for 24 hours under both acidic (pH 3) and alkaline (pH 11) conditions, and constant permeance ($0.155 \pm 0.005 \text{ L m}^{-2} \text{ h}^{-1} \text{ bar}^{-1}$) through the membrane (35nm/XP12-D) was observed in all pH conditions while rejecting the permeation of the dye molecules. The dye rejection was 97.90 %, 98.67 %, and 97.97 % for neutral, acidic, and alkaline conditions, respectively (**Figure for Review**). These results strongly support the environmental stability of the membrane under harsh chemical environments.

4. The permeance of these membranes is quite low even with 30 nm thick selective layers. What strategies could be employed to improve the permeance of these membranes towards viable levels (~10 LMH/bar)?

We strongly agree with the Reviewer that the strategy for enhancing the permeance of the initiated chemical vapor deposition (iCVD) membranes is critically important for practical applications. In this manuscript, we focused on the synthesis of ultrathin iCVD membranes and their applications in highly selective OSN. For further improvement, we would provide our strategy on iCVD membranes from two perspectives: 1) a versatile combination of chemical functionalities for membrane synthesis with enhanced permeance, and 2) highly selective OSN membranes for the purification of real-world APIs.

In this study, we fabricated an ultrathin, highly interconnected organosiloxane membrane via the iCVD process. The same can be applied to the synthesis of copolymer film – a combination of two different kinds of chemical functionalities. The vapor-phase polymer synthesis method enables conformal coverage deposition of an ultrathin copolymeric nanolayer on various substrates through the homogeneous mixing of vaporized monomer species, regardless of their physical and chemical properties. The resultant nanoscale copolymer film features a uniform and homogeneous copolymer

composition. The rich history of generating copolymeric nanolayers on electronic or biogenic devices through the iCVD technique has established a wide range of vaporizable monomer combinations, including long linear alkyl chains, bulky aromatic rings, or other cross-linkable functional groups. (1) These unique advantageous characteristics enable facile copolymer deposition through a myriad of combinations using various monomers more readily than the conventional solution-based approach. This iCVD strategy has been proven effective in various other applications, including electronic devices and the biomedical field (2,3,4). For example, we can start from the current membrane design with highly interconnected pV4D4 homopolymer film. The free volume of the highly interconnected pV4D4 can be enlarged by copolymerizing the V4D4 monomer with other monomers with bulky side chains and, thus, higher free volume. With such a strategy, we believe we can systematically adjust the free volume of the polymer film, and we hope we can maximize the permeance without mitigating the selectivity of the V4D4 framework. Indeed, we are currently employing cross-linked copolymer polymer film in order to further improve the OSN performance of the membrane, which we hope we can present in the near future as a separate study.

Our current manuscript discussed the highly interconnected network of an organosilicon polymer with exceptionally high selectivity for solutes with low solvent permeance. Especially our membrane shows very high “solute-solute selectivity” for a mixture of solute pairs with slightly different size from each other; for example, one solute in 150 – 250 g mol⁻¹ and the other solute in 250 – 350 g mol⁻¹ can be separated with sharp contrast, verified by polystyrene rejection in the original manuscript.

For further employment of the pV4D4 membrane, we attempted to employ excellent membrane selectivity in real-world API purification in the revised manuscript, highlighting the selectivity of the iCVD membranes. Zovirax® (Acyclovir, MW: 225.20 g mol⁻¹), a market product of GlaxoSmithKline, is widely used as an antiviral drug for treating the Herpes virus. However, due to its relatively low

bioavailability, various prodrugs, such as ValtrexTM (Valacyclovir, MW: 324.34 g mol⁻¹), famciclovir (MW: 321.33 g mol⁻¹), and penciclovir (MW: 253.26 g mol⁻¹), have been synthesized from Acyclovir. To enhance the medicinal efficacy of the target drug, the purity of the product must be maximized, requesting precise separation of these APIs after the synthesis of prodrugs. To illustrate the effectiveness of the pV4D4 membrane in purifying API products, the separation of Acyclovir (MW: 225.20 g mol⁻¹) and Valacyclovir (MW: 324.34 g mol⁻¹) was attempted with 29nm/XP12-D membrane. The API separation experiment clearly confirmed the highly promising performance of the iCVD membranes with acyclovir/valacyclovir selectivity exceeding 11 (newly added **Figure 5e** in the revised manuscript). While the iCVD membranes indeed show low solvent permeance compared to other conventional membranes, we demonstrated the precision separation of API with unprecedentedly high selectivity that had never been achieved from polymer-based membrane, is accomplished with iCVD membranes even with only a MW difference of less than 100 g mol⁻¹.

We added the following description, statements, and new results in the main text:

- **Page 4, Lines 71:** Zovirax® (Acyclovir, MW: 225.20 g mol⁻¹), a market product of GlaxoSmithKline, is widely used as an antiviral drug for treating the Herpes virus. However, due to relatively low bioavailability of Acyclovir (15 – 30%), various prodrugs of Acyclovir, such as ValtrexTM (Valacyclovir, MW: 324.34 g mol⁻¹), famciclovir (MW: 321.33 g mol⁻¹), and penciclovir (MW: 253.26 g mol⁻¹) have been designed from the Acyclovir to enhance its medicinal effects.^[9-11] To enhance medicinal efficacy, the purity of the product should be high, which requires precise separation of these APIs after the synthesis.
- **Page 18-19, Lines 379:** To demonstrate the effectiveness of the 29nm/XP12-D membrane

in purifying API products, we further evaluated the separation of Acyclovir (MW: 225.20 g mol⁻¹) and Valacyclovir (MW: 324.34 g mol⁻¹) in methanol. This solute pair (i.e., solute-solute) matches the molecular weight cutoff range provided by 29nm/XP12-D that solute molecules in 150 – 250 g mol⁻¹ (i.e., Acyclovir) can be separated from the solute molecules in 250 – 350 g mol⁻¹ (i.e., Valacyclovir). The permeation of Acyclovir and Valacyclovir through 29nm/XP12-D were tested with methanol, and HPLC results showed retention times of 4.8 minutes and 8.6 minutes for Acyclovir and Valacyclovir, respectively (**Figure 5e**). The split peak observed in the permeate at 4.4 minutes was due to the formation of a potential intermediate of Acyclovir in the HPLC solvents.^[49] The calculated rejections were 88.95 ± 3.26 % and 99.00 ± 0.53 % for Acyclovir and Valacyclovir, respectively. Despite a molecular weight difference of less than 100 g mol⁻¹ between Acyclovir and Valacyclovir, the unprecedentedly high “solute-solute” selectivity between these solutes was achieved as 11.04, providing the precision API separation that has previously been unattainable in polymer-based membranes.

- **Page 20, Lines 418:** With this emerging process, copolymerization using a wide range of monomers given in the iCVD libraries will facilitate the elaborate design of next-generation high-performance iCVD membranes.^[52,53]
- **Page 22, Lines 463:** The analysis of API products (Acyclovir and Valacyclovir) was performed using the same conditions as the polystyrene analysis, with the exception that the mobile phase consisted of 95 vol % water, 5 vol % acetonitrile, and 0.19 vol % formic acid.

■ **Figure 5e** was newly added:

Figure 5. (e) Ultra-violet absorbance of API products (Acyclovir and Valacyclovir) was measured by HPLC at both and permeates of 29nm/XP12-D, corresponding to retention time.

The following references in the main body were modified:

- [52] K. Jeong, D. Kim, D. Ahn, C. Yang, J. Kim, C. Lee, Y. Kim, C. Lee, Y.-S. Park, S.-H. Lee, T.-S. Kim, S. G. Im, *Chem. Eng. J.* **2024**, *480*, 148151.
- [53] J. Ryu, M. S. Oh, J. Yoon, M. Kang, J. B. You, H. Lee, S. G. Im, *J. Mater. Chem. C* **2023**, *11* (13), 4318-4327.

References used in this comment:

- (1) Gleason, K. K. Designing Organic and Hybrid Surfaces and Devices with Initiated Chemical Vapor Deposition (iCVD). *Adv. Mater.* **2023**, 2306665.
- (2) Oh, J.; Kim, S.; Lee, C.; Cha, J.-H.; Yang, S. Y.; Im, S. G.; Park, C.; Jang, B. C.; Choi, S.-Y. Preventing Vanishing Gradient Problem of Hardware Neuromorphic System by Implementing Imidazole-Based Memristive ReLU Activation Neuron. *Adv. Mater.* **2023**, *35* (24), 2300023.
- (3) Bae, J.; Choi, K.; Song, H.; Kim, D. H.; Youn, D. Y.; Cho, S.-H.; Jeon, D.; Lee, J.; Lee, J.;

Jang, W.; Lee, C.; Kim, Y.; Kim, C.; Jung, J.-W.; Im, S. G.; Kim, I.-D. Reinforcing Native Solid-electrolyte Interphase Layers via Electrolyte-swellable Soft-scaffold for Lithium Metal Anode. *Adv. Energy Mater.* **2023**, *13* (16), 2203818

- (4) Chen, P.; Lang, J.; Zhou, Y.; Khlyustova, A.; Zhang, Z.; Ma, X.; Liu, S.; Cheng, Y.; Yang, R. An Imidazolium-Based Zwitterionic Polymer for Antiviral and Antibacterial Dual Functional Coatings. *Sci Adv* **2022**, *8* (2), eabl8812.

5. The polystyrene solution used for rejection testing was fairly dilute in concentration. Were any other solutes of higher concentration tested with these membranes? Do you maintain high rejection under more concentrated feed conditions?

As the Reviewer pointed out, the concentration of the polystyrene solution (0.8 g L^{-1}) used in this work was not relatively high. We employed an equal-weight mixture of three PS oligomer standard reference materials (each 0.1 g L^{-1}) with a polydispersity index (PDI) of 1.00 (nominal MW: 162, 266, and 370 g mol^{-1}) and one PS mixture (0.5 g L^{-1}) (nominal MW: 750 g mol^{-1}) with a polydispersity index (PDI) of 1.09. Using the monodisperse PS oligomer solution led us to determine the solute rejection in the 162 – 370 g mol^{-1} range with a much higher cutoff resolution than just using a PS mixture with a higher PDI. Meanwhile, mixing PS oligomers and PS mixtures could affect the peak splitting in the subsequent HPLC measurement using a UV-Vis detector. The optimal PS concentration we settled was around 0.8 g L^{-1} . The works from previous literature used only one PS oligomer (α -methylstyrene dimer, MW: 236 g mol^{-1}) and PS mixtures to test solute rejection, which we believe are not enough to demonstrate solute-solute selectivity in the low molecular weight range (1,2).

To address the Reviewer's concern about the solute concentration, we additionally carried out an OSN test under a higher concentration of other solutes (see **Figure for Review**). We prepared a methylene blue (MB) ($319.85 \text{ g mol}^{-1}$) in a methanol feed solution with a concentration of 30 mg L^{-1} , far exceeding the conventional laboratory-tested concentration of $10 - 20 \text{ mg L}^{-1}$ found in previous

Figure for Review. Methylene blue (MW: 319.85 g mol⁻¹) rejection at dye concentrations of 15 mg L⁻¹ and 30 mg L⁻¹.

literature (3,4,5). The OSN test clearly demonstrated that more than 97% of dye rejection was achieved from the high-concentration (30 mg L⁻¹) using 35nm/XP12-D, similar to the control experiment using a low-concentration (15 mg L⁻¹) MB solution – 97.90 % for 30 mg L⁻¹ and 98.53 % for 15 mg L⁻¹, respectively. This observation confirms the membrane performance can be retained in a wide range of feed concentrations.

References used in this comment:

- (1) Rivera, M. P.; Bruno, N. C.; Finn, M. G.; Lively, R. P. Organic Solvent Reverse Osmosis Using CuAAC-Crosslinked Molecularly-Mixed Composite Membranes. *J. Memb. Sci.* **2021**, *638*, 119700.
- (2) Cook, M.; Gaffney, P. R. J.; Peeva, L. G.; Livingston, A. G. Roll-to-Roll Dip Coating of Three Different PIMs for Organic Solvent Nanofiltration. *J. Memb. Sci.* **2018**, *558*, 52–63.
- (3) Huang, T.; Moosa, B. A.; Hoang, P.; Liu, J.; Chisca, S.; Zhang, G.; AlYami, M.; Khashab, N. M.; Nunes, S. P. Molecularly-Porous Ultrathin Membranes for Highly Selective Organic Solvent Nanofiltration. *Nat. Commun.* **2020**, *11* (1), 5882.
- (4) Zhang, Y.; Kim, D.; Dong, R.; Feng, X.; Osuji, C. O. Tunable Organic Solvent Nanofiltration in Self-Assembled Membranes at the Sub-1 Nm Scale. *Sci Adv* **2022**, *8* (11), eabm5899.
- (5) Jimenez-Solomon, M. F.; Song, Q.; Jelfs, K. E.; Munoz-Ibanez, M.; Livingston, A. G.

Polymer Nanofilms with Enhanced Microporosity by Interfacial Polymerization. *Nat. Mater.* **2016**, *15* (7), 760–767.

6. What is the fouling/regeneration behavior of these membranes when they are exposed to higher concentration solutions?

The Reviewer pointed out a critical aspect of practical membrane separation operation. Following the high-concentration OSN test from comment #5, we further investigated the fouling and regeneration behavior of the pV4D4 membrane. After the dye permeation test was completed, membranes were removed from the testing cell and visually inspected. The optical images (see **Figure for Review**) of the 29nm/XP12-D membrane showed a partial change in surface color from white (pristine) to blue, indicating that the surface/subsurface was contaminated due to organic dye permeation. Subsequently, the tested membrane was immersed in DMF for 1 hour to mimic a regeneration step for removing organic foulant. As can be seen in the optical image below, the DMF treatment successfully removed most of the organic foulant, recovering the white pristine color of the original membrane. In our membrane preparation method in the main text, we employ the “DMF activation” step to prepare the iCVD membrane (**Figure 4b**), and we believe this simple treatment can also be used to regenerate the

Figure for Review. Optical images of the 29nm/XP12-D membrane (a) before and (b) after DMF regeneration.

membrane when fouling occurs.

7. Can you comment on the mechanical properties of the membranes before and after iCVD deposition? Does the addition of the highly cross-linked siloxane significantly decrease the membrane flexibility?

Thank you for the critical comments. The mechanical properties of the highly interconnected siloxane membrane should be discussed in the manuscript, providing insights into membrane handling. From the perspective of handling the membrane, our material design for selective layer incorporates the utilization of highly interconnected pV4D4 with extraordinarily flexible -Si-O-Si- bonding not to compromise the mechanical properties of the entire membrane but rather to complement them. For instance, the ultrathin pV4D4 interlayer can play a pivotal role in enhancing interfacial adhesion and even imparting stretchable properties (1). As the Reviewer might have anticipated, most of the mechanical properties of the pV4D4/XP membrane would stem from the mechanical properties of the support membrane since the volumetric fraction of the iCVD layer (29 nm-thick) is extremely small – practically negligible (see **Figure for Review**). In this work, 220 μm -thick support XP membranes were prepared on the polypropylene/polyethylene non-woven fabric. Therefore, the mechanical behavior of the membrane

Figure for Review. Simplified illustration of the pV4D4/XP membrane structure.

(pV4D4 + XP + fabric) would be practically the same as that of the non-woven fabric, which shows excellent mechanical flexibility. To prove this point, we conducted tensile-strength tests (new **Supplementary Figure 17**) of the composite membrane (29nm/XP12) and the support membrane (XP12). The stress-strain curves of XP12 and 29nm/XP12 membranes were monitored by applying different stresses to the membrane. As expected, both membranes showed nearly identical responses under mechanical deformation, and the overall mechanical flexibility was dominated by the support membrane.

The following parenthetical and data were added to the main text.

- **Page 13, Lines 271:** Notwithstanding the highly interconnected nature of the ultrathin membrane layer, the pV4D4/XP membrane shows no noticeable compromise in mechanical flexibility (**Supplementary Figure 17**).
- The following **Supplementary Fig. 17**, was added.

Supplementary Fig. 17. Stress-strain curve of membrane tensile testing results. All tensile test was performed using a universal testing machine (5kN AllRound table top, ZwickRoell, Germany). The film specimens were prepared with the following dimensions: 10 mm in width, 50 mm in length, and 220 μm in thickness. The film membranes were pulled at strain rate of 20 mm min^{-1} at room temperature, starting from the initial grip gap of 10 mm.

References used in this comment:

- (1) Oh, M. S.; Ryu, J.; Jeon, M.; Lee, I.; Bae, B.-S.; You, J. B.; Im, S. G. A Fully Transparent, Stretchable Multi-layered Water Barrier Thin Film for the Passivation of Underwater Device Applications. *Adv. Mater. Interfaces* **2022**, *9*, 2201019.

8. Do these membranes undergo loss of permeance after aging?

As the Reviewer pointed out, many polymeric membranes, especially with high fractional free volume, would undergo aging and generally show reduced permeance due to the loss of fractional free volume over time. Since the iCVD membranes prepared in this study are highly interconnected, we expected that the polymer aging process related to the free volume reduction could be minimized in our membranes. We newly attempted a longer-term dye permeation test for an extended 10 days to investigate the possible aging behavior of 35nm/XP12-D membranes. We used methylene blue (MW: 319.85 g mol⁻¹) in methanol at a relatively high dye concentration of 30 mg L⁻¹. As can be seen from

Figure for Review. Long-term methanol permeance of 35nm/XP12-D membranes for aging behavior.

the **Figure for Review** below, except for the slight fluctuation in MeOH permeance at the initial phase due to the start-up of an experiment, the membrane performance remained constant for ten consecutive days. It is noteworthy that the statistical errors in permeance are largely decreased over time, which indicates the steady-state membrane flux is achieved in a 10-day period. Again, this result is attributed to the unique chemical structure of the highly interconnected pV4D4, which possesses a low free volume.

We added the following statements in the main text:

- **Page 16-17, Lines 342-344:** Resistance to time, pressure, and harsh solvents was speculated to be conferred by the unique chemical structure of the highly interconnected pV4D4, which possesses low free volume.

Reviewer #2

Remarks to the Author: The manuscript by Dong-Yeun Koh and coworkers reports an ultrathin, cross-linked organosiloxane membrane for precision organic solvent nanofiltration. The fabrication of organosiloxane membrane using CVD method has been reported (J. Membr. Sci. 230 (2004) 49–59). The development of this work is using (1,3,5,7-tetravinyl-1,3,5,7-tetramethylcyclotetrasiloxane) (pV4D4) as the precursor. However, the advantage of the new precursor is not fully presented. The membrane separation performance is not comparable with the reported polyamide membrane. The manuscript doesn't meet the standards that make it improper to publish in Nature Communications.

First of all, thank you for reviewing our manuscript thoroughly and providing many important and critical comments. The Reviewer mentioned that the use of the iCVD membrane was not a novel approach and questioned its advantages for organic solvent nanofiltration. We made every effort to address all the issues raised by the Reviewer properly. Before providing detailed responses and new additional results on the membrane performances in the subsequent section, we would like to clarify our motivation for using the iCVD process and its novelty in membrane fabrication.

As the Reviewer has pointed out, the use of the conventional CVD method for OSN membrane fabrication has already been reported by several researchers. One of the CVD methods, the plasma-enhanced chemical vapor deposition (PECVD) process, has been utilized to create thin membranes used for organic solvent nanofiltration (OSN), among other membrane applications. However, the PECVD process is often associated with a myriad of uncontrollable side reactions of organic reactants due to the excessively high energy in the plasma phase. Such inherent uncontrollable nature of the PECVD polymer films cannot warrant a sufficient level of reliability of the membrane performance, thus critically hampering its application to the membranes requiring a high degree of stability (1). Unlike the

high-energy plasma-based methods, the iCVD process is a plasma-free method for synthesizing various functional polymers under mild process conditions of ambient temperature (20 – 50 °C) without the use of solvent. This mild condition provides a low-energy landscape of the reactants, minimizing the damage in the functional groups (e.g., cross-linkable sites) of the monomer units during the polymerization (2). iCVD process fabricates homogeneous membrane structures with no side reactions, leading to the high usage of cross-linkable points in the monomer reactant. These inherent advantages of the iCVD process allowed us to fabricate ultrathin (usually less than 100 nm) but highly interconnected, and thus chemically stable membranes with minimal defect sites that could not have been processed in other conventional membrane fabrication methods such as non-solvent induced phase separation (NIPS) or thermal induced phase separation (TIPS). Such highly interconnected ultrathin polymers with full retention of their important chemical functionalities are difficult to generate via a conventional solution process. Notably, unlike traditional membranes, this low-free-volume membrane via the iCVD process can overcome the challenging membrane stability issues, including aging, compaction, and long-term operation in the harsh solvents required for OSN. In the vapor-phase technique, homogeneous mixing and injection ratio control of vaporized precursors can facilitate precise control over the polymer structure–inner pores and their distribution—and the capability offers high selectivity in distinguishing subtle differences in molecular weights of desired substances.

References used in this comment:

- (1) Trujillo, N. J.; Baxamusa, S. H.; Gleason, K. K. Grafted Functional Polymer Nanostructures Patterned Bottom-Up by Colloidal Lithography and Initiated Chemical Vapor Deposition (iCVD). *Chem. Mater.* **2009**, *21* (4), 742–750.
- (2) Tenhaeff, W. E.; Gleason, K. K. Initiated and Oxidative Chemical Vapor Deposition of Polymeric Thin Films: iCVD and oCVD. *Adv. Funct. Mater.* **2008**, *18* (7), 979–992.

1. The author mentioned that the resulting membranes were heavily cross-linked membranes, how do we define them ‘heavily’? There is no specific characterization to certify high cross-linking degree of the material in comparison with other polymers, and it is also lack of corresponding experiments to regulate the cross-linking of the membrane. Besides, the author also emphasizes the high-density of the membrane, the relationship between the two concepts should be clearly expounded.

Thank you very much for your considerate comments. We fully acknowledge that we indeed used the term “heavily cross-linked” in a confusing manner. In the original manuscript, we used the phrase “heavily cross-linked” to the pV4D4 without proper assessment of its cross-linking density. When we designed the iCVD-based OSN membrane, we selected the V4D4 monomer because one V4D4 monomer contains many cross-linkable vinyl groups – 4 in this case, and we simply assumed the monomer with many cross-linkable sites would form a “heavily cross-linked” polymer network when synthesized as pV4D4 polymer film. The high degree of consumption of vinyl moiety in V4D4 monomer during the iCVD process was evidently confirmed by FT-IR spectra analysis. (**Figure 3c**) Given that the cross-linked network of pV4D4 is directly associated with the vinyl group-mediated free-radical polymerization reaction, we can assume that the degree of vinyl group consumption in pV4D4 can serve as a good indicator to estimate the relative level of cross-linking of pV4D4. Indeed, the degree of vinyl group consumption in pV4D4 can be controlled systematically in the iCVD process simply by tuning the input monomer-to-initiator feed ratios (see new **Supplementary Figure 15**). When comparing the intensity of the vinyl groups in FT-IR spectra (i.e., 1598 cm^{-1}), higher monomer-to-initiator ratios resulted in the higher peak intensity representing the C=C vibration in FT-IR spectra, indicating that a considerable amount of residual vinyl group survived the vapor-phase free radical polymerization step, and thereby strongly inferring that the resultant pV4D4 is “less” cross-linked.

Hence, the phrase “the heavily cross-linked” may lead to misunderstanding or open interpretation from which we overhauled the controversial phrase in the context.

On the other hand, we fully agree with the Reviewer that comparing the density of organosilicon pV4D4 with other Si-free polymers may be inadequate due to the higher atomic mass of Si than other atoms such as C or N. We had no intention of addressing the complex correlation between density and degree of cross-linking, and it is not feasible in iCVD membranes. As demonstrated in various literature, all authors are well aware that the degree of network cross-linking in polyamide membranes can be calculated through XPS elemental ratios (1). In the case of polyamide, the fabrication of a nanolayer proceeds from the condensation polymerization using heterogeneous precursors, i.e., diamine and terephthalic acid, containing different elements and its skeletal framework varies depending on the ratio of precursors, from which the degree of cross-linking of polyamide membrane can be contingent on elemental ratio proved by spectroscopic investigations.

In contrast, the pV4D4 is prepared through the radical polymerization of a single V4D4 monomer, and the conversion of the vinyl group to ethylene in XPS shows no change in the elemental composition. Even with the high-resolution carbon peaks in XPS spectra, quantifying the subtle transition from C – C to C = C is challenging. Moreover, the radical polymerization can lead to the unlinked vinyl groups of V4D4 due to the increased local steric hindrance during synthesis, adding complexity to the spectroscopic analysis. Given the intricate structure of the pV4D4, it seems prudent to refrain from discussing the degree of cross-linking in a direct manner.

Additionally, the refractive index of the pV4D4 films increases as the monomer-to-initiator ratio decreases, and the higher value usually indicates a highly interconnected structure (2,3). The measured refractive index values of pV4D4 films ($n = 1.49 \sim 1.50$) were higher than that of other conventional organosiloxane polymer, i.e., poly(dimethylsiloxane) (PDMS) ($n = 1.41 \sim 1.43$), indicating a higher

cross-linking density (not degree of cross-linking).

For clarity, we revised the title, words, main body, and Supplementary Table 3 as follows:

- **Title:** Ultrathin Organosiloxane Membrane for Precision Organic Solvent Nanofiltration
- **Words:** Replace the words from ‘heavily cross-linked’ to ‘highly interconnected’
- **Page 13, Lines 267:** When compared to the densities of other silicone polymers, such as poly(dimethylsiloxane) (PDMS) (0.97 g cm^{-3}) or 1,2-bis(triethoxysilyl)ethane (BTESE) (1.50 g cm^{-3}), this high-density value indicates the formation of a more highly interconnected pV4D4 network than other silicone polymers (**Supplementary Table 3**).^[41,42]
- **Page 12-13, Lines 256:** To investigate the cross-linking structure of pV4D4, various monomer-to-initiator flow ratios were adjusted. When checking the peak intensity variation representing C=C vibration in FT-IR spectra (1598 cm^{-1}), lower monomer-to-initiator ratios resulted in the lower peak intensity, indicating that a considerable amount of vinyl group participated in the vapor-phase free radical polymerization step, and thereby strongly inferring that the resultant pV4D4 is highly interconnected (**Supplementary Figure 15a,b**). Additionally, the refractive index of the pV4D4 films showed an increasing trend as the monomer-to-initiator ratios decreased (**Supplementary Figure 15c**). The calculated refractive index of pV4D4 films ($n = 1.49 \sim 1.50$) was higher than that of other typical organosiloxane polymer, poly(dimethylsiloxane) (PDMS) ($n = 1.41 \sim 1.43$), indicating a higher cross-linking density.^[36-40]

Supplementary Table 3. was modified:

Supplementary Table 3. Bulk density of polymers in comparison with other silicone polymers.

Silicone polymer	Density (g cm ⁻³)	Reference
PDMS*	0.97	[2]
BTESE*	1.50	[3]
pV4D4	1.70	This work

* Abbreviations: PDMS-poly(dimethylsiloxane), BTESE-1,2-bis(triethoxysilyl)ethane

The **Supplementary Fig. 15.** was modified.

Supplementary Fig. 15. pV4D4 cross-linking structure analysis. (a-b) FT-IR spectra and (c) refractive index of pV4D4 films with varying monomer-to-initiator ratios.

The following references were modified.

- [36] V. Prajzler, P. Nekvindova, J. Spirkova, M. Novotny, *J. Mater. Sci.: Mater. Electron.* **2017**, *28*, 7951.
- [37] V. Gupta, P. T. Probst, F. R. Goßler, A. M. Steiner, J. Schubert, Y. Brasse, T. A. F. König, A. Fery, *ACS Appl. Mater. Interfaces* **2019**, *11*, 28189.
- [38] Y. Su, E. B. D. S. Filho, N. Peek, B. Chen, A. E. Stiegman, *Macromolecules* **2019**, *52*, 9012.
- [39] S. D. Bhagat, J. Chatterjee, B. Chen, A. E. Stiegman, *Macromolecules* **2012**, *45*, 1174.
- [40] L. Y. Tyng, M. R. Ramli, M. B. H. Othman, R. Ramli, Z. A. M. Ishak, Z. Ahmad, *Polym. Int.* **2013**, *62*, 382.
- [41] Y. Yamada, T. Ichii, T. Utsunomiya, H. Sugimura, *Jpn. J. Appl. Phys. (2008)* **2020**, *59*, SN1009.
- [42] L. Cui, A. N. Ranade, M. A. Matos, L. S. Pingree, T. J. Frot, G. Dubois, R. H.

Reference used in this comment:

- (1) Karan, S.; Jiang, Z.; Livingston, A. G. Sub-10 Nm Polyamide Nanofilms with Ultrafast Solvent Transport for Molecular Separation. *Science* **2015**, *348* (6241), 1347–1351.
- (2) Higashihara, T.; Ueda, M. Recent Progress in High Refractive Index Polymers. *Macromolecules* **2015**, *48* (7), 1915–1929.
- (3) Jang, W.; Choi, K.; Kang, M.; Park, S.; Kim, D. H.; Ahn, J.; Lim, H.; Char, K.; Lim, J.; Im, S. G. Visible, Mid- and Long-Wave Infrared Transparent Sulfur-Rich Polymer with Enhanced Thermal Stability. *Chem. Mater.* **2023**, *35* (19), 8181–8191.

2. The membrane showed the selectivity of the molecule with the MWCO of 150-300 and 250-300 g mol⁻¹, but the significance of molecular sieve fractions at this scale is not clearly illustrated. In addition, IP is a widely investigated process of NF and RO membrane fabrication, the thickness, cross-linking degree and screening performance of the selective layer can be well regulated in previous reports. The IP based membrane with ~10 nm thickness and low MWCO of ~200Da with high permeability have been reported in previous studies. In this work, the membrane shows the lower flux with similar MWCO value, the advantages of the iCVD are not well demonstrated which makes this work less innovative.

The Reviewer raises an excellent point. This study focuses on the highly interconnected polymer network that offers outstanding selectivity toward solutes despite low solvent permeability. Our membrane displays remarkable “solute-solute selectivity” specifically for separating solute molecules in 150 – 250 g mol⁻¹ from other solute molecules in 250 – 350 g mol⁻¹, as indicated by the polystyrene rejection in the original manuscript. Numerous high-value active pharmaceutical ingredients (APIs) are present in this MW range, which could benefit from less energy-intensive membrane processes. We were intrigued by the Reviewer’s comments regarding the selectivity and permeability of the iCVD membranes. To probe this question more deeply, we experimentally attempted to translate membrane

selectivity into real-world API purification, highlighting the selectivity of iCVD membranes.

Zovirax® (Acyclovir, molecular weight (MW): 225.20 g mol⁻¹), manufactured by GlaxoSmithKline, is a commonly prescribed antiviral medication for the treatment of the Herpes virus. Yet, because of its limited ability to be absorbed and utilized by the body, several prodrugs, including Valtrex™ (Valacyclovir, MW: 324.34 g mol⁻¹), famciclovir (MW: 321.33 g mol⁻¹), and penciclovir (MW: 253.26 g mol⁻¹), have been chemically created from Acyclovir. In order to optimize medical effectiveness, it is crucial to ensure a high purity level in the product. This necessitates the precision separation of the (APIs) following their synthesis. The successful use of the pV4D4 membrane in purifying API products is demonstrated here, specifically in separating Acyclovir (MW: 225.20 g mol⁻¹) and Valacyclovir (MW: 324.34 g mol⁻¹) with 29nm/XP12-D membrane. This solute pair (i.e., solute-solute) exactly matches the molecular weight cutoff range provided in the original manuscript that solute molecules in 150 – 250 g mol⁻¹ (Acyclovir) can be separated from the solute molecules in 250 – 350 g mol⁻¹ (Valacyclovir). The API separation experiment further confirmed the remarkable selectivity of the iCVD membranes, as illustrated by the Acyclovir/Valacyclovir selectivity above 11 (as shown in the new **Figure 5e**). Although the iCVD membranes have lower solvent permeance compared to other conventional membranes, we have successfully demonstrated that precise separation of APIs can still be achieved in iCVD membranes, even when the molecular weight difference between solute molecules is less than 100 g mol⁻¹ (324.34 g mol⁻¹ vs. 225.20 g mol⁻¹).

Furthermore, as the Reviewer mentioned, the interfacial polymerization (IP) based polymer membranes have high solvent permeance and low MWCO. The key distinction for iCVD membrane lies in emphasizing solute-solute separation rather than solute-solvent separation, as demonstrated clearly by the API purification experiment. Still, we also fully acknowledge the Reviewer's concern that the permeance enhancement in these membranes is critical to compete with other IP-based

membranes. The vapor-phase synthesis method affords conformal coverage of an ultrathin polymeric nanolayer on various substrates through the homogeneous mixing of vaporized precursors, regardless of their physical and chemical properties (1,2,3). The resultant nanoscale film features a uniform structure and homogeneous backbone composition. The rich history of making polymeric nanolayers on electronic or biogenic devices through the iCVD technique has established a wide range of vaporizable monomer libraries, including long linear alkyl chains, bulky aromatic rings, or other cross-linkable functional groups (4). These unique processing advantages enable facile copolymer deposition through a myriad of combinations using various monomers more readily than the conventional solution-based approach. For example, we can start from the current membrane design with highly interconnected pV4D4 homopolymer film. The free volume of the highly interconnected pV4D4 can be enlarged by copolymerizing the V4D4 monomer with other monomer with bulky side chain, and thus higher free volume. With such strategy we believe we can systematically adjust the free volume of the polymer film, and we hope we can maximize the permeance without mitigating the selectivity of the V4D4 framework. Indeed, we are currently employing cross-linked copolymer polymer film in order to further improve the OSN performance of the membrane, which we hope we can present in near future as a separate study.

We added the following description, statements, and new results in the main text:

- **Page 4, Lines 71:** Zovirax® (Acyclovir, MW: 225.20 g mol⁻¹), a market product of GlaxoSmithKline, is widely used as an antiviral drug for treating the Herpes virus. However, due to relatively low bioavailability of Acyclovir (15 – 30 %), various prodrugs of Acyclovir, such as Valtrex™ (Valacyclovir, MW: 324.34 g mol⁻¹), famciclovir (MW: 321.33 g mol⁻¹), and penciclovir (MW: 253.26 g mol⁻¹), have been designed from the Acyclovir to enhance its medicinal effects.^[9-11] To enhance medicinal efficacy, the purity

of the product should be high, which requires precise separation of these APIs after the synthesis.

- **Page 18-19, Lines 379:** To demonstrate the effectiveness of the 29nm/XPI2-D membrane in purifying API products, we further evaluated the separation of Acyclovir (MW: 225.20 g mol⁻¹) and Valacyclovir (MW: 324.34 g mol⁻¹) in methanol. This solute pair (i.e., solute-solute) matches the molecular weight cutoff range provided by 29nm/XPI2-D that solute molecules in 150 – 250 g mol⁻¹ (i.e., Acyclovir) can be separated from the solute molecules in 250 – 350 g mol⁻¹ (i.e., Valacyclovir). The permeation of Acyclovir and Valacyclovir through 29nm/XPI2-D were tested with methanol, and HPLC results showed retention times of 4.8 minutes and 8.6 minutes for Acyclovir and Valacyclovir, respectively (**Figure 5e**). The split peak observed in the permeate at 4.4 minutes was due to the formation of a potential intermediate of Acyclovir in the HPLC solvents.^[49] The calculated rejections were 88.95 ± 3.26 % and 99.00 ± 0.53 % for Acyclovir and Valacyclovir, respectively. Despite a molecular weight difference of less than 100 g mol⁻¹ between Acyclovir and Valacyclovir, the unprecedentedly high “solute-solute” selectivity between these solutes was achieved as 11.04, providing the precision API separation that has previously been unattainable in polymer-based membranes.
- **Page 20, Lines 418:** With this emerging process, copolymerization using a wide range of monomers given in the iCVD libraries will facilitate the elaborate design of next-generation high-performance iCVD membranes.^[52,53]
- **Page 22, Lines 471:** The analysis of API products (Acyclovir and Valacyclovir) was performed using the same conditions as the polystyrene analysis, with the exception that the mobile phase consisted of 95 vol % water, 5 vol % acetonitrile, and 0.19 vol % formic

acid.

- **Figure 5e** was newly added:

Figure 5. (e) Ultra-violet absorbance of API products (Acyclovir and Valacyclovir) was measured by HPLC at both and permeates of 29nm/XP12-D, corresponding to retention time.

The following references in the main body were added/modified:

- [9] C. Gege, F. J. Bravo, N. Uhlig, T. Hagmaier, R. Schmachtenberg, J. Elis, A. Burger-Kentischer, D. Finkelmeier, K. Hamprecht, T. Grunwald, D. I. Bernstein, G. Kleymann, *Sci. Transl. Med.* **2021**, *13*, eabf8668.
- [10] O. Y. Mady, S. M. Thabit, S. E. Abo Elnasr, A. A. Hedaya, *Sci. Rep.* **2023**, *13*, 20067.
- [11] S. Jain, K. Ansari, S. Maddala, S. Meenakshisunderam, *World Patent* **2017**.
- [49] B. C. Bejgum, P. R. Johnson, W. C. Stagner, *Int. J. Pharm.* **2018**, *535*, 172.
- [52] K. Jeong, D. Kim, D. Ahn, C. Yang, J. Kim, C. Lee, Y. Kim, C. Lee, Y.-S. Park, S.-H. Lee, T.-S. Kim, S. G. Im, *Chem. Eng. J.* **2024**, *480*, 148151.
- [53] J. Ryu, M. S. Oh, J. Yoon, M. Kang, J. B. You, H. Lee, S. G. Im, *J. Mater. Chem. C* **2023**, *11* (13), 4318-4327.

References used in this comment:

- (1) Asatekin, A.; Gleason, K. K. Polymeric Nanopore Membranes for Hydrophobicity-Based Separations by Conformal Initiated Chemical Vapor Deposition. *Nano Lett.* **2011**, *11* (2), 677–686.
- (2) Servi, A. T.; Guillen-Burrieza, E.; Warsinger, D. M.; Livernois, W.; Notarangelo, K.; Kharraz, J.; Lienhard, J. H., V.; Arafat, H. A.; Gleason, K. K. The Effects of iCVD Film Thickness and Conformality on the Permeability and Wetting of MD Membranes. *J. Memb. Sci.* **2017**, *523*, 470–479.
- (3) Joo, M.; Shin, J.; Kim, J.; You, J. B.; Yoo, Y.; Kwak, M. J.; Oh, M. S.; Im, S. G. One-Step Synthesis of Cross-Linked Ionic Polymer Thin Films in Vapor Phase and Its Application to an Oil/Water Separation Membrane. *J. Am. Chem. Soc.* **2017**, *139* (6), 2329–2337.
- (4) Gleason, K. K. Designing Organic and Hybrid Surfaces and Devices with Initiated Chemical Vapor Deposition (iCVD). *Adv. Mater.* **2023**, e2306665.

3. According to Figure 5, the support substrate will affect the PS rejection, so which functional layer ultimately determines the selectivity of the membrane?

We appreciate the Reviewer for raising a critical point. The support substrates do not show any appreciable solute selectivity in the suggested MWCO range. Solvent permeance values through the support substrates (PAN or XP) are described in **Supplementary Figure 2c**, where order-of-magnitude greater solvent permeances were observed than those of composite membranes with pV4D4/XP. We fabricated a PAN support substrate and subsequently cross-linked the PAN to improve the solvent resistance of the membranes. The cross-linking step would induce the formation of a tighter membrane structure, showing that the surface terminal pore size was decreased considerably in the XP6 membrane (18 nm of PAN to 14 nm of XP6) (**Supplementary Figure 2b**). As the Reviewer pointed out, membrane performance data in **Figure 5** describes how these surface pores in the support substrates affect the pV4D4 layer performance on OSN. The solvent permeance values in **Supplementary Figure 2c** indicate that the solvent permeance of the cross-linked PAN membrane decreased as the cross-linking time increased from 6 hours to 12 hours, meaning that extended cross-linking reaction induced

a tighter membrane structure with a smaller surface pore size. The smaller surface pores would also reduce the surface roughness and decrease the penetration of monomer vapors into the subsurface during the iCVD process. The membrane performance data in **Figure 5** indicates that better-performing pV4D4 membranes are produced on the porous PAN substrate with tighter, smaller pore structures. Additionally, as the thickness of the pV4D4 layer increased, we observed a remarkable enhancement in the rejection of solutes due to the formation of a highly interconnected structure originating from pV4D4. Considering that the rejection for solutes with smaller molecular weights increases significantly with a slight increase in the thickness of the pV4D4 layer, it can be inferred that the pV4D4 top layer mainly determines the selectivity of the membrane.

The following sentences were added and modified in the manuscript:

- **Page 17, Lines 359:** These large differences in PS rejection profiles highlighted the importance of smaller surface pores, which reduce surface roughness and the diffusive penetration of monomer vapors into the substrate during the iCVD process.
- **Page 18, Lines 366:** As the thickness of the pV4D4 layer increased to 35 nm, a remarkable enhancement in the rejection of solutes was obtained due to the formation of a tighter structure (**Figure 5c**).

4. Meanwhile, the manuscript mentioned the resulting membrane was an ultrathin membrane, but the methanol permeability of the membrane is very low, and I did not see any highlight performance in the organic solvent nanofiltration.

Thank you for the suggestion. We believe this comment is connected with comment #2. Please refer to

our response to comment #2. As described in comment #2, our highly interconnected pV4D4 membrane demonstrated a distinct strength in “solute-solute” separation, which is demanded by high-value industries rather than solute-solvent separation. In contrast, solute-solvent separation might be constrained by limitations in achieving solute-solute selectivity due to concentration polarization, which can be caused by the permeation of a large amount of solvent, making it unsuitable for high-value industries.

5. How to tune the pore size as the author mentioned in Abstract? Detailed experimental results are needed?

We apologize for providing confusing information in the Abstract. In our work, although we did not adjust the pore size of the organosiloxane polymer, we controlled the pore size of the XP membrane substrate by varying the cross-linking time (please refer to response to comment #3). The sentence with ambiguous expression was revised as follows:

- **Page 1-2, Lines 24:** We optimize the iCVD poly(1,3,5,7-tetravinyl-1,3,5,7-tetramethylcyclotetrasiloxane) (pV4D4) membrane by adjusting both the thickness of the selective layer and the pore sizes of its support membranes.

Furthermore, as we provided a detailed strategy to enhance the permeance of the iCVD membranes in response to comment #2, copolymerization is a feasible approach in iCVD to enhance the membrane performance.

6. This sentence “the 29nm pV4D4 nanoscale selective layer fabricated in the vapor phase provides a homogeneous, narrow molecular sieving property, enabling a record-high solute selectivity of 39.88 in

the range of between 150 – 250 g mol⁻¹ and 250 – 350 g mol⁻¹.” is unclear in expression. Why are the two ranges involved?

Thank you for pointing this out. We apologize for the unclear sentence. What we intended in the original manuscript was that we could separate two different-sized solutes, one in 150 – 250 g mol⁻¹ and the other in 250 – 350 g mol⁻¹, through the pV4D4 membrane. As the Reviewer noted, we demonstrated real-world API purification (Acyclovir and Valacyclovir) to prove this statement above.

Also, we modified several sentences in the manuscript for a clear description of our intentions:

- **Page 2, Lines 30:** Notably, the 29 nm pV4D4 selective layer imparts a uniformly narrow molecular sieving property, providing a record-high “solute-solute selectivity” of 39.88 for two different-sized solutes; the separation between small solute (in 150 – 250 g mol⁻¹) and larger solute (in 250 – 350 g mol⁻¹) is achievable. Furthermore, a solute-solute selectivity of 11.04 was demonstrated using the real-world active pharmaceutical ingredient mixture of Acyclovir and Valacyclovir, key components for Herpes virus treatment, despite their molecular weight difference of less than 100 g mol⁻¹.

7. Why the membrane needs to undergo a 24-hour restructuring in DMF?

In the process of fabricating the pV4D4 membrane, we found that the presence of unreacted monomers and oligomers negatively impacts permeance. Therefore, when we conducted soaking or permeation tests in various solvents, we found that restructuring (permeation at 30 bar pressure) in DMF solvent for 24 hours improved permeance substantially. After 24 hours of continued permeation, the DMF flux

reached a constant value, leading us to judge it as a steady state (**Figure 4e**). Accordingly, all rejection data in the manuscript were measured after undergoing the 24-hour restructuring process in DMF.

The following sentence was modified in the manuscript:

- **Page 2, Lines 26:** The resulting pV4D4 membrane undergoes restructuring in dimethylformamide solvent for 24 hours to ensure sufficient porosity and remove unreacted monomers and oligomers.

8. During the iCVD process, the unreacted monomers are easy to deposit on the substrate membrane. In this work, the authors tried to remove the monomers and by-products by DMF treatment. However, during the separation process, the flux of DMF increases with increasing the filtration time which is different from other solvents. Whether it is due to the membrane swelling or incompletely removal of by-products?

Membranes subjected to DMF treatment are designated with a "D" suffix in their nomenclature. The results from unmarked membranes were demonstrated only to assess the usefulness of the DMF-based restructuring process. **Figure 4a** shows the permeation of several solvents in series using a single membrane for the entire experimental campaign (data was triplicated with different membranes). In the change of solvent flux with the sequential solvent test, we observed clearly that drastically improved solvent permeance of the membrane after DMF permeation. The origin of this effect is mainly discussed as DMF restructuring in the original manuscript, where the removal of unreacted V4D4 monomers or oligomeric V4D4 in the substrate region significantly decreased the mass transfer resistance in the composite membrane structure. However, the DMF only shows a tendency to increment in permeance

over time and this tendency is not observed in other solvents. (**Figure 4a**) All the solvents tested in the sequential permeation test show a time-dependent increase in permeance; for example, MeOH showed a transient increase of permeance at a time marked before 100 hours and reached a steady state after the time marked after 100 hours. We tested all membrane permeation until the membranes reached a steady state. The steady-state permeances were achieved in other solvent sequential tests shown in **Figure 4b**; please refer to the MeOH permeance in the 5th and 7th sequence, showing nearly identical permeance. Based on these observations, we could assure that all our membrane performances reported in the manuscript are obtained in steady states and not under the influence of the transient swelling or remaining impurities in the membrane.

The following sentences were modified in the manuscript:

- **Page 15, Lines 318:** After 24 hours, the permeating solvent was changed to acetone, and the permeance through the same membrane also showed a slight increase over time.
- **Page 16, Lines 325:** The enhanced permeance effect by the DMF treatment, compared to other solvents, aligned well with the depth analysis result that the DMF activation step would remove unreacted monomers and oligomers in the V4D4-percolated region, as observed by ToF-SIMS depth profiles.

Reviewer #3

Remarks to the Author:

This work reports a highly selective organosiloxane membrane prepared by iCVD for organic solvent nanofiltration (OSN) applications. As the authors highlighted, the precise separation of small molecules with an Mw range of 150–300 g/mol is very important, which has been overlooked by many studies. Overall, I felt the authors could successfully address the main challenge in the OSN field, supported by their novel ideas and solutions. I recommend its publication after considering my several comments to improve this work.

1. The authors claim that the prepared pV4D4 membranes are “heavily” cross-linked, but I could not find the rationale behind the term “heavily”. In other words, have the authors estimated the cross-linking density of the pV4D4 membranes? Specifically, I was concerned about the presence of a C=C peak in pV4D4 (Fig. 3c), which indicates that the presence of unreacted monomer is still significant. I understand that the cross-linking density estimation is very challenging for thin films, so I recommend the authors use an alternate expression throughout the paper (for example, just “ultrathin organosiloxane membrane” would be fine in the title).

Thank you for pointing this out. In the original manuscript, we used the phrase “heavily cross-linked” based on comparing pV4D4 with other similar silicon-based polymers with the intention of highlighting its higher density due to a more significant number of cross-linkable points in the pV4D4 monomer. However, we fully agree with the Reviewer that the rationale for the “heavily” word and comparing the density with polymers that do not contain Si atoms is inadequate due to its higher atomic mass than other atoms such as C or N.

For clarity, we revised the title as follows:

- **Title: Ultrathin Organosiloxane Membrane for Precision Organic Solvent Nanofiltration**

2. Can the authors explain more details about the density estimation by XRR? For example, is it bulk density or true density?

Thank you for the suggestion. Per to your comment, we modified the manuscript and supplemented more details about the density estimation by XRR analysis.

In the Supporting Information,

X-ray Reflectometry (XRR) analysis

The XRR spectrum was obtained using a Rigaku SmartLab X-ray diffractometer using X-ray with a wavelength of 1.541 Å. The bulk density of the pV4D4 film deposited on Si wafer was estimated by fitting XRR measurement and simulation values, executed using the GlobalFit software.

For further understanding, **Supplementary Text** is added to demonstrate more details about density estimation on the XRR analysis as below:

Supplementary Text

In the XRR analysis, the bulk density of thin film can be estimated by determining the critical angle, θ_c , for total reflection.^[1] The X-ray refractive index of a material is represented in a complex number (Eq. 1) and is slightly less than 1 as a result of X-ray surface scattering and adsorption, where δ depends on the wavelength-dependent scattering associated with density and composition of the material (Eq. 2) and β is influenced by the X-ray adsorption (Eq. 3). The value of δ is on

the order of 10^{-6} for X-ray with a wavelength of approximately $\sim 1 \text{ \AA}$, but that of β is nearly negligible, on the order of 10^{-8} .

$$n = 1 - \delta - i\beta \quad (1)$$

$$\delta = \left(\frac{r_e \lambda^2}{2\pi} \right) N_0 \rho \sum_i x_i \left(z_i + f_i' \right) / \sum_i x_i M_i \quad (2)$$

$$\beta = \left(\frac{r_e \lambda^2}{2\pi} \right) N_0 \rho \sum_i x_i \left(z_i + f_i'' \right) / \sum_i x_i M_i \quad (3)$$

r_e : Classical radius of an electron ($2.818 \times 10^{-9} \text{ m}$)

N_0 : Avogadro number

λ : X-ray wavelength

ρ : density (g cm^{-3})

z_i : Atomic number of the i^{th} atom

M_i : Atomic weight of the i^{th} atom

x_i : Atomic ratio (molar ratio) of the i^{th} atom

f_i', f_i'' : Atomic scattering factors of the i^{th} atom (anomalous dispersion term)

When the incident angle of X-ray is tilted to the critical angle in an initial state where the X-ray beam is nearly parallel to the film surface, total reflection will occur. Since the refraction angle is 90° at the total reflection, Eq. 1 can be re-expressed as Eq. 4 using Snell's Law. As shown in Eq. 5, the monitored θ_c of the thin film depends on the δ value including the density information of the substance. This allows for the estimation of bulk density through the obtained θ_c for total reflection.

$$n \approx 1 - \delta, 1 - \delta = \cos \theta_c \approx 1 - \frac{\theta_c^2}{2} \quad (4)$$

$$\theta_c \approx \sqrt{2\delta} \quad (5)$$

In our work, the θ_c for the pV4D4 thin film is 0.37° , and the eventual bulk density is estimated to be 1.70 g cm^{-3} .

3. I felt more details about the structure-property-performance relationship are needed. The authors only characterized d -spacing as a potential contributor to the OSN performance. Is there any other way to estimate the structural properties of the pV4D4 materials (e.g., BET?).

We fully understand your concern about the structure-property-performance relationship of the pV4D4 materials. However, obtaining sufficient measurement specimens for analyzing properties like BET surface, as mentioned by the Reviewer, was challenging due to the quite low deposition rate (1.41 nm min^{-1}) of pV4D4. Instead, for indirect estimation of cross-linked structure, we newly prepared pV4D4 films synthesized with different ratios of monomer-to-initiator in the iCVD chamber to study the changes in vinyl groups and density (see new **Supplementary Figure 15**). The change in the monomer-to-initiator ratio is critically linked to the interconnected structure of pV4D4. When comparing the intensity of the vinyl groups in FT-IR spectra (i.e., 1598 cm^{-1}), lower monomer-to-initiator ratios resulted in the lower peak intensity, indicating that a considerable amount of the vinyl group participated in the vapor-phase free radical polymerization step, and thereby strongly inferring that the resultant pV4D4 is highly interconnected. Additionally, the refractive index of the pV4D4 films also increases as the monomer-to-initiator ratio decreases, this higher value indicates a denser cross-linked structure (1,2). The measured refractive index values of pV4D4 films ($n = 1.49 \sim 1.50$) were higher than that of the typical organosiloxane polymer, i.e., poly(dimethylsiloxane) (PDMS) ($n = 1.41 \sim 1.43$), indicating a higher cross-linking density (not degree of cross-linking).

We added the following sentences to the main body of the manuscript:

- **Page 12-13, Lines 256:** To investigate the cross-linking structure of pV4D4, various monomer-to-initiator flow ratios were adjusted. When checking the peak intensity variation representing C=C vibration in FT-IR spectra (1598 cm^{-1}), lower monomer-to-initiator ratios resulted in the lower peak intensity, indicating that a considerable amount of vinyl group participated in the vapor-phase free radical polymerization step, and thereby strongly inferring that the resultant pV4D4 is highly interconnected (**Supplementary Figure 15a,b**). Additionally, the refractive index of the pV4D4 films showed an increasing trend as the monomer-to-initiator ratio decreased (**Supplementary Figure 15c**). The calculated refractive index of pV4D4 films ($n = 1.49 \sim 1.50$) was higher than that of other typical organosiloxane polymer, poly(dimethylsiloxane) (PDMS) ($n = 1.41 \sim 1.43$), indicating a higher cross-linking density.^[36-40]
- The **Supplementary Fig. 15** was modified.

Supplementary Fig. 15. pV4D4 cross-linking structure analysis. (a-b) FT-IR spectra and (c) refractive index of pV4D4 films with varying monomer-to-initiator ratios.

The following references were modified.

- [36] V. Prajzler, P. Nekvindova, J. Spirikova, M. Novotny, *J. Mater. Sci.: Mater. Electron.* **2017**, *28*, 7951.
- [37] V. Gupta, P. T. Probst, F. R. Gößler, A. M. Steiner, J. Schubert, Y. Brasse, T. A. F. König, A. Fery, *ACS Appl. Mater. Interfaces* **2019**, *11*, 28189.

- [38] Y. Su, E. B. D. S. Filho, N. Peek, B. Chen, A. E. Stiegman, *Macromolecules* **2019**, *52*, 9012.
- [39] S. D. Bhagat, J. Chatterjee, B. Chen, A. E. Stiegman, *Macromolecules* **2012**, *45*, 1174.
- [40] L. Y. Tyng, M. R. Ramli, M. B. H. Othman, R. Ramli, Z. A. M. Ishak, Z. Ahmad, *Polym. Int.* **2013**, *62*, 382.

Reference used in this comment:

- (1) Higashihara, T.; Ueda, M. Recent Progress in High Refractive Index Polymers. *Macromolecules* **2015**, *48* (7), 1915–1929.
- (2) Jang, W.; Choi, K.; Kang, M.; Park, S.; Kim, D. H.; Ahn, J.; Lim, H.; Char, K.; Lim, J.; Im, S. G. Visible, Mid- and Long-Wave Infrared Transparent Sulfur-Rich Polymer with Enhanced Thermal Stability. *Chem. Mater.* **2023**, *35* (19), 8181–8191.

4. In Fig. 5, it is interesting to observe the subtle change in membrane thickness (only ~6 nm) between 29nm/XP12-D and 35nm/XP12-D resulting in a huge difference in the rejection profiles. Could the authors comment on the reason behind this behavior?

Thank you for suggesting a deeper explanation of the differences in the rejection profiles. As pointed out by the Reviewer, a notable difference in rejection profiles was observed clearly between the 29nm/XP12-D and 35nm/XP12-D membranes. As the thickness of the pV4D4 layer increased, we observed a remarkable enhancement in the rejection of solutes due to the formation of a tighter structure.

The following sentences were added in the manuscript:

- **Page 18, Lines 366:** As the thickness of the pV4D4 layer increased to 35 nm, a remarkable enhancement in the rejection of solutes was obtained due to the formation of a tighter structure (**Figure 5c**).

5. Can the authors provide more details on the polystyrene rejection estimation? For example, some people use solvent transfer to achieve distinguishable HPLC peaks.

As mentioned by the Reviewer, we utilized the ethanol-based solvent transfer method to achieve clear HPLC peaks for polystyrene with a molecular weight of 266 g mol⁻¹ or above. However, the polystyrene HPLC peak with a molecular weight of 162 g mol⁻¹ was volatile and directly analyzed without further treatment.

The following description was added to the methods section of the main text.

- **Page 22, Lines 454-458:** To ensure distinct HPLC peaks for polystyrenes with a molecular weight of 266 g mol⁻¹ or above, we first evaporated all solvents and subsequently re-dissolved them in ethanol before HPLC analysis.^[58] However, for polystyrene with a molecular weight of 162 g mol⁻¹, due to its volatile nature, we analyzed it directly without any treatment.

The following new reference was added.

- **[58] M. P. Rivera, N. C. Bruno, M. G. Finn, R. P. Lively, *J. Memb. Sci.* **2021**, 638, 119700.**

6. For GIWAXS analysis, the authors mentioned it was measured using a pV4D4 film coated onto a silicon substrate. How did the authors prepare the sample? Did the authors transfer the film from the XP substrate to the silicon substrate?

Thank you for pointing this out. To examine the intrinsic properties of pV4D4, we directly deposited the pV4D4 layer on a silicon wafer substrate (Grazing Incidence Wide Angle X-ray Scattering (GIWAXS) analysis Method section in Supporting Information). Unfortunately, separating only the

pV4D4 layer from the XP substrate is at least extremely challenging, if not impossible, because the superior adhesion between the selective layer and support was achieved via the penetration of monomer vapor into the PAN membrane substrate during the iCVD process, evidenced by the ToF-SIMS analysis results in the manuscript. Furthermore, GIWAXS relies on very low-angle light (almost parallel to the surface) for the structure analysis of the thin film sample, necessitating a very low surface roughness. Considering all these aspects, we performed GIWAXS analysis for the pV4D4 layer by depositing it on a flat silicon wafer rather than on a membrane surface.

7. Is there any theoretical basis for the proposed new upper bound in Fig. 5d?

Thank you for pointing this out. For gas separation, the Robeson plot illustrates the trade-off relationship between the permeability and selectivity of polymer membranes and best describes upper-bound performance. This upper-bound relation can be derived from an empirical basis for many gas pairs using published permeability and selectivity data through the solution-diffusion model (1). However, the organic solvent nanofiltration does not fit well with the solution-diffusion model, and due to the extremely diverse operating conditions from many reported performance data, a theoretical basis for the upper bound has not yet been established, to the best of our knowledge. Therefore, the upper bound line was derived by statistical assessment of data points in many other previous literatures to illustrate a trade-off relationship between solvent permeance and solute rejection (2,3,4,5).

- (1) Freeman, B. D. Basis of Permeability/Selectivity Trade-off Relations in Polymeric Gas Separation Membranes. *Macromolecules* **1999**, *32* (2), 375–380.
- (2) Jimenez-Solomon, M. F.; Song, Q.; Jelfs, K. E.; Munoz-Ibanez, M.; Livingston, A. G. Polymer Nanofilms with Enhanced Microporosity by Interfacial Polymerization. *Nat. Mater.* **2016**, *15* (7), 760–767.
- (3) Jiang, Z.; Dong, R.; Evans, A. M.; Biere, N.; Ebrahim, M. A.; Li, S.; Anselmetti, D.; Dichtel, W. R.; Livingston, A. G. Aligned Macrocyclic Pores in Ultrathin Films for

Accurate Molecular Sieving. *Nature* **2022**, 609 (7925), 58–64.

- (4) Liang, B.; Wang, H.; Shi, X.; Shen, B.; He, X.; Ghazi, Z. A.; Khan, N. A.; Sin, H.; Khattak, A. M.; Li, L.; Tang, Z. Microporous Membranes Comprising Conjugated Polymers with Rigid Backbones Enable Ultrafast Organic-Solvent Nanofiltration. *Nat. Chem.* **2018**, 10 (9), 961–967.
- (5) Li, S.; Dong, R.; Musteata, V.-E.; Kim, J.; Rangnekar, N. D.; Johnson, J. R.; Marshall, B. D.; Chisca, S.; Xu, J.; Hoy, S.; McCool, B. A.; Nunes, S. P.; Jiang, Z.; Livingston, A. G. Hydrophobic Polyamide Nanofilms Provide Rapid Transport for Crude Oil Separation. *Science* **2022**, 377 (6614), 1555–1561.

8. The represented density of polyimide (1.4 g/cm^3) can be significantly varied depending on which type of polyimide (high or low free volume). Please consider it.

As the Reviewer pointed out, we did not mention the polyimide type for density comparison in the original manuscript. However, comparing the density with polymers that do not contain Si atoms may not be appropriate due to its higher atomic mass than other atoms, such as C or N.

For clarity, we revised the following sentences to the main body of the manuscript:

- **Page 13, Lines 267:** When compared to the densities of other silicone polymers, such as poly(dimethylsiloxane) (PDMS) (0.97 g cm^{-3}) or 1,2-bis(triethoxysilyl)ethane (BTESE) (1.50 g cm^{-3}), this high-density value indicates the formation of a more highly interconnected pV4D4 network than other silicone polymers (**Supplementary Table 3**).^[41,42]

Supplementary Table 3, was modified:

Supplementary Table 3. Bulk density of polymers in comparison with other silicone polymers.

Silicone polymer	Density (g cm ⁻³)	Reference
PDMS*	0.97	[2]
BTESE*	1.50	[3]
pV4D4	1.70	This work

* Abbreviations: PDMS-poly(dimethylsiloxane), BTESE-1,2-bis(triethoxysilyl)ethane

The following references in the main body were modified:

- [41] Y. Yamada, T. Ichii, T. Utsunomiya, H. Sugimura, *Jpn. J. Appl. Phys. (2008)* **2020**, 59, SN1009.
- [42] L. Cui, A. N. Ranade, M. A. Matos, L. S. Pingree, T. J. Frot, G. Dubois, R. H. Dauskardt, *ACS Appl. Mater. Interfaces* **2012**, 4, 6587.

9. The designation of sample names is quite confusing. For example, I assume that 178 nm/XP6 means that 178 nm of pV4D4 was coated onto the XP6 membrane, but no definition was provided in the main text. In this sense, what is the sample shown in Fig. S7a?

Thank you for the suggestion. We included additional details about the membrane nomenclature in the main body.

- **Page 6, Lines 125-127:** The prepared pV4D4 membranes were termed based on the measured membrane thickness (in nanometers), the type of support membrane (XP6 or XP12, numeric values represent cross-linking time), and the DMF activation.

As previously mentioned, measuring membrane thickness is essential for naming the membrane. However, we did not measure the membrane thickness used in Fig. S7a (pV4D4/PAN SEM image in

original manuscript). Therefore, since Fig. S7a was intended to visually represent the V4D4-percolated region, we replaced it with EDS mapping images to avoid confusion regarding the membrane name.

Therefore, the following statement was modified to the main body.

- **Page 9, Lines 185:** The cross-sectional EDS mapping images of the 29nm/XP12 membrane clearly illustrated the presence of the V4D4-percolated region (**Supplementary Figure 7**).
- **Supplementary Fig. 7.** has been added.

Supplementary Fig. 7. Cross-sectional SEM image and elemental mapping images of 29nm/XP12 membrane. (a) Cross-sectional SEM image of the 29nm/XP12 membrane, along with (b-d) elemental mapping images of carbon, oxygen, and silicon, respectively. Scale bar: 3 μm

Reviewers' Comments:

Reviewer #1:

Remarks to the Author:

The authors thoroughly and comprehensively addressed the many comments to the initial manuscript. I believe the additional experimental evidence and clarification to the text, especially the new separation of API compounds, provides clear evidence to support their claims. This work is an important new addition to the OSN literature and helps to address many of the challenges in differentiating between very small MW molecules in organic solvents and should be published in Nature Communications.

Reviewer #2:

Remarks to the Author:

The authors have satisfactorily addressed all my concerns. The manuscript is suitable for publication.

Reviewer #3:

Remarks to the Author:

The authors have successfully addressed my comments and I feel the paper became much more consistent and impactful after this revision. I am happy to accept this paper for the publication in Nature Communications.

Point-by-Point Responses

Reviewer #1 (Remarks to the Author):

The authors thoroughly and comprehensively addressed the many comments to the initial manuscript. I believe the additional experimental evidence and clarification to the text, especially the new separation of API compounds, provides clear evidence to support their claims. This work is an important new addition to the OSN literature and helps to address many of the challenges in differentiating between very small MW molecules in organic solvents and should be published in Nature Communications.

Your feedback was really beneficial in enhancing and clarifying our manuscript. We appreciate your time to review our manuscript.

Reviewer #2 (Remarks to the Author):

The authors have satisfactorily addressed all my concerns. The manuscript is suitable for publication.

Your recommendations and inquiries have provided us with the opportunity to add scientific depth to our work.

Reviewer #3 (Remarks to the Author):

The authors have successfully addressed my comments and I feel the paper became much more consistent and impactful after this revision. I am happy to accept this paper for the publication in Nature Communications.

Your comments and suggestions helped us to revise our manuscript. We appreciate your helpful comments.